# Out-of-Distribution Detection with An Adaptive Likelihood Ratio on Informative Hierarchical VAE

**Yewen Li**[1]* **Chaojie Wang**[1]*† **Xiaobo Xia**[2] **Tongliang Liu**[2] **Xin Miao**[3] **Bo An**[1]
[1]Nanyang Technological University [2]University of Sydney [3]Amazon

## Abstract

Unsupervised out-of-distribution (OOD) detection is essential for the reliability of machine learning. In the literature, existing work has shown that higher-level semantics captured by hierarchical VAEs can be used to detect OOD instances. However, we empirically show that, the inheirt "*posterior collapse*" of hierarchical VAEs would seriously limit their capacity for OOD detection. Based on a thorough analysis, we propose an informative hierarchical VAE to alleviate this issue through enhancing the connections between the data sample and its multi-layer stochastic latent representations during training. Furthermore, we propose a novel score function for unsupervised OOD detection, referred to as Adaptive Likelihood Ratio, which can selectively aggregate the semantic information on multiple hidden layers of hierarchical VAEs, leading to a strong separability between in-distribution and OOD samples. Experimental results demonstrate that our method can significantly outperform existing state-of-the-art unsupervised OOD detection approaches.

## 1 Introduction

Despite achieving great success in real-world applications recently, existing machine learning (ML) systems are still designed to be tested on the in-distribution dataset, whose statistics are similar to those of the training set [1]. However, when applied to deal with the dataset consisting of out-of-distribution (OOD) samples, whose statistics are extremely different from those of the training set, these ML systems would produce a series of incorrect judgments [2, 3, 4]. Considering the fact that OOD data is very common in real-world applications, pre-execution OOD detection is increasingly attractive to make sure the reliability and safety of ML systems. Although several supervised methods [5, 6, 7, 8, 9, 10, 11, 12, 13, 14, 15, 16] have achieved great success in OOD detection, the unsupervised ones are more practical since the category labels of in-distribution samples are often missing in real-world applications, which brings more challenges for OOD detection and is also the focus of this work.

Without labels, likelihood-based models could be a promising way for unsupervised OOD detection, such as flow-based models [17], auto-regressive models [18, 19], and variational autoencoders (VAEs) [20, 21, 22, 21, 23]. Unfortunately, some recent studies have shown that, in some cases, these generative models tend to achieve higher likelihoods on certain types of OOD samples [1, 24, 25, 26, 27], which makes the OOD detection methods based on thresholding likelihood scores problematic. To address this issue, based on the prior knowledge collected from OOD samples, some improvements have been made for unsupervised OOD detection, *i.e.*, Ren et al. [1] take additional datasets as the OOD validation sets for choosing the best hyperparameters; Hendrycks et al. [28] introduce an auxiliary dataset to teach the network to learn more expressive representations for OOD detection; However, the behaviour of borrowing the prior knowledge of OOD data is usually

---

*Equal contributions.
†Corresponding to: Chaojie Wang <chaojie.wang@ntu.edu.sg>.

36th Conference on Neural Information Processing Systems (NeurIPS 2022).

unreasonable in practice, because we will never know the statistic information of OOD samples to be dealt with.

To conduct purely unsupervised OOD detection without the labels or prior assumptions, deep ensemble method named WAIC [24] is developed by making full use of the difference between the density estimations of multiple independent models trained on the in-distribution data. Recently, through capturing the semantic information with multi-layer latent variables, Maaløe et al. [29] and Havtorn et al. [27] develop various score functions based on Likelihood-Ratio, which help hierarchical VAEs achieve competitive performance in unsupervised OOD detection. However, in our implementation, we find that the phenomenon of "*posterior collapse*" in hierarchical VAEs still limits their performance on OOD detection, where the main reason could be that "*posterior collapse*" will make the high-level latent variables meaningless and cannot provide faithful summaries for the data.

With this insight in hand, in this paper, we start from rethinking the cause of "*posterior collapse*" in hierarchical VAEs, and then theoretically explain why "*posterior collapse*" will limit the OOD detection performance of these Likelihood-Ratio based methods. Further, we develop an informative hierarchical VAE to alleviate "*posterior collapse*" and a novel Adaptive Likelihood Ratio score function for unsupervised OOD detection. The major contributions of this work include:

- With a thorough analysis of "*posterior collapse*" in hierarchical VAEs, we enhance the connections between the data sample and its multiple latent representations in the expected log-likelihood term of evidence lower bound (ELBO) for training, and develop a novel informative hierarchical VAE to extract more expressive hierarchical latent representations.
- We theoretically explain why alleviating "*posterior collapse*" in hierarchical VAEs can help the performance of Likelihood Ratio on OOD detection, and then develop a novel score function for fully unsupervised OOD detection, termed Adaptive Likelihood Ratio, which owns fewer hyperparameters to be tuned and can make full use of the semantic divergences between in-distribution and OOD samples across all hidden layers of hierarchical VAEs.
- Combing the informative hierarchical VAE with the Adaptive Likelihood Ratio, we demonstrate that our method can achieve state-of-the-art OOD detection performance across a wide range of benchmarks in an unsupervised manner.

## 2 Background and Related Works

### 2.1 Hierarchical Variational Autoencoder

**Preliminary:** Extending the basic VAE [20], a hierarchical VAE [30, 31] is defined by the observation $\boldsymbol{x}$ that depends on a hierarchy of stochastic latent variables $\boldsymbol{z} = \boldsymbol{z}_1, ..., \boldsymbol{z}_L$, where the generative model is defined with a top-down structure, formulated as $p_\theta(\boldsymbol{x}, \boldsymbol{z}) = p_\theta(\boldsymbol{x}|\boldsymbol{z}_1) \prod_{l=1}^{L-1} p_\theta(\boldsymbol{z}_l|\boldsymbol{z}_{l+1}) p_\theta(\boldsymbol{z}_L)$; the inference model is designed to approximate the posterior over these latent variables, commonly factorized with a top-down structure as $q_\phi(\boldsymbol{z}|\boldsymbol{x}) = \prod_{l=1}^{L-1} q_\phi(\boldsymbol{z}_l|\boldsymbol{z}_{l+1}) q_\phi(\boldsymbol{z}_L|\boldsymbol{x})$ or a bottom-up structure as $q_\phi(\boldsymbol{z}|\boldsymbol{x}) = \prod_{l=1}^{L-1} q_\phi(\boldsymbol{z}_{l+1}|\boldsymbol{z}_l) q_\phi(\boldsymbol{z}_1|\boldsymbol{x})$. The demonstration of these structures could be seen in Fig. 1. The parameters of the generative and infernce models, denoted as $\theta$ and $\phi$ respectively, can be jointly optimized by maximizing the evidence lower bound (ELBO) expressed as

$$\mathcal{L} = \mathbb{E}_{p(\boldsymbol{x})} \left[ \mathcal{L}_x(\boldsymbol{x}; \theta, \phi) \right], \tag{1}$$

where $\mathcal{L}_x$ is denoted as

$$\mathcal{L}_x = \log p(\boldsymbol{x}) - D_{\mathrm{KL}}(q_\phi(\boldsymbol{z}|\boldsymbol{x}) || p_\theta(\boldsymbol{z}|\boldsymbol{x})) = \mathbb{E}_{q_\phi(\boldsymbol{z}|\boldsymbol{x})} \left[ \log p_\theta(\boldsymbol{x}|\boldsymbol{z}) \right] - D_{\mathrm{KL}}(q_\phi(\boldsymbol{z}|\boldsymbol{x}) || p_\theta(\boldsymbol{z})), \tag{2}$$

where $D_{\mathrm{KL}}(\cdot || \cdot)$ denotes the KL divergence and maximizing ELBO is equivalent to minimize the divergence between the varaitional distribution $q_\phi(\boldsymbol{z}|\boldsymbol{x})$ and true posterior $p_\theta(\boldsymbol{z}|\boldsymbol{x})$.

**Related Works:** While the hierarchy of latent stochastic variables can improve the generation capability of standard VAEs, in practice, the posterior of higher-level stochastic latent variables have a tendency to *collapse* into the prior, called "*posterior collapse*". To address this issue, Sønderby et al. [30] propose a Ladder VAE (LVAE) to change the bottom up inference process into a top-down one; Vahdat and Kautz [31] develop a variant of the LVAE, which carefully designs a sophisticated network architecture to achieve better generation quality; Maaløe et al. [29] combine the bottom-up inference with the top-down inference by proposing a bidirectional inference scheme. Despite obtaining performance improvements with more flexible inference networks, these hierarchical VAEs still lack a theoretical guide to alleviate the phenomenon of "*posterior collapse*".

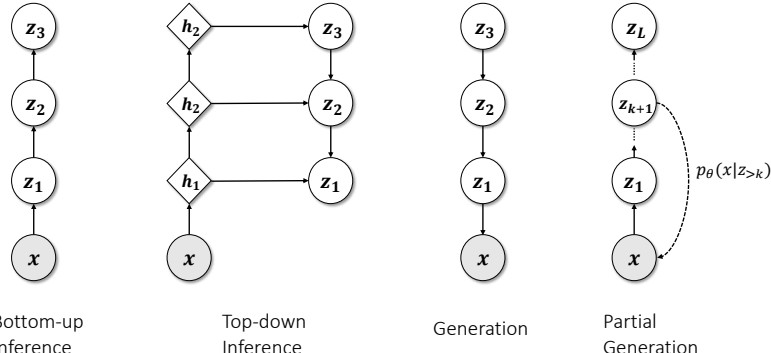

Figure 1: Illustration of the usual structures of both inference network and generative model in hierarchical VAEs.

## 2.2 Out-of-distribution Detection with Variational Autoencdoer

**Problem Formulation:** Suppose that there are a set of $N$ training samples $\{x_i\}_{i=1}^N$ drawn from the data distribution $x_i \sim p(x)$, after training a VAE to model the generation of these data samples, the generative model $p_\theta(x)$ is supposed to detect whether a testing sample $x$ is an outlier, where the outlier should have a low density estimation under the true data distribution $p(x)$. However, on the contrary, former likelihood-based methods found the generative model always assign higher $p_\theta(x)$ for OOD data than in-distribution data [1, 24, 25, 26]. Luckily, although the likelihood based methods with VAE are rarely investigated, they have revealed the potential to better address this problem without the help of labels or assumptions about the prior.

**Related Works:** A pioneering VAE-based OOD detection method is Likelihood Regret (LRe) [32] obtained by iteratively finetuning the decoder parameters of VAE, which is time-consuming but achieves competitive performance in an unsupervised manner. Maaløe et al. [29] reveal the potential of hierarchical VAE for OOD detection and Havtorn et al. [27] further propose a score function to improve the model performance, which needs to specify the backbone model. The backbone models in likelihood-based methods [1, 24, 33] can be directly replaced with VAEs, but these methods usually underperform flow-based models like Glow [17] or autoregressive models like PixelCNN [19].

## 3 From Informative Hierarchical VAE to Adaptive Likelihood Ratio

### 3.1 Rethinking of "*posterior collapse*" in Hierarchical VAEs

Firstly, let's understand the cause of "*posterior collapse*" in hierarchical VAEs theoretically. Taking an $L$-layer hierarchical VAE with a top-down inference network as an example, the set of latent variables can be separated as the lower-level variables $z_{\leq k} = \{z_1, ..., z_k\}$ and the higher-level ones $z_{>k} = \{z_{k+1}, ..., z_L\}$, where $k \in \{0, ..., L-1\}$, then the ELBO in Eq. (1) can be reformulated as

$$\mathcal{L} = \mathbb{E}_{p(x)} \left[ \mathbb{E}_{q_\phi(z_{\leq k}|z_{>k})q_\phi(z_{>k}|x)} \left[ \log p_\theta(x|z_1) \right] - \sum_{l=1}^{L} D_{\text{KL}}(q_\phi(z_l|z_{l+1})||p_\theta(z_l|z_{l+1})) \right], \quad (3)$$

where $q_\phi(z_L|z_{L+1}) := q_\phi(z_L|x)$, $p_\theta(z_L|z_{L+1}) := p_\theta(z_L)$, and the main contribution to the expected log-likelihood term is coming from the lower-level latent variables $z_{\leq k}$ before the $k$th hidden layer [29]. Once the generation capacity of the generative model $p_\theta(x|z_{\leq k})$ is powerful enough to reconstruct the observation $x$ well, the variational posteriors of higher-level latent variables $z_{>k}$ will be optimized to be close to their priors, i.e., $q_\phi(z_{>k}|x) \approx p_\theta(z_{>k})$, leading the representations learned by VAE at higher layers to be meaningless and cannot provide faithful summaries for $x$, which is well-known as the phenomenon of "*posterior collapse*" or "*latent variable collapse*" [30, 34].

To find the potential solutions to alleviating "*posterior collapse*", in the following, we reinterpret this phenomenon from the perspective of information theory [35] by extending the findings in [34] to a hierarchical VAE scenario. For ease of undertanding, we define the mutual information between the data $x$ and the higher-level latent variables $z_{>k}$ as

$$\mathcal{I}_q(x, z_{>k}) = -\mathcal{H}_q(z_{>k}|x) + \mathcal{H}_q(z_{>k}) = \mathbb{E}_{p(x)q_\phi(z_{>k}|x)} \log q_\phi(z_{>k}|x) - \mathbb{E}_{q_\phi(z_{>k})} \log q_\phi(z_{>k}),$$

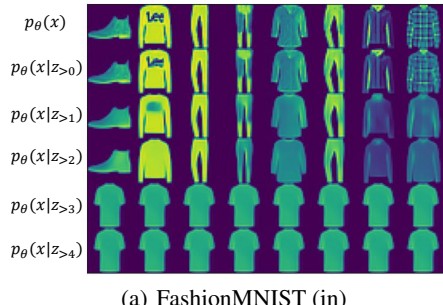

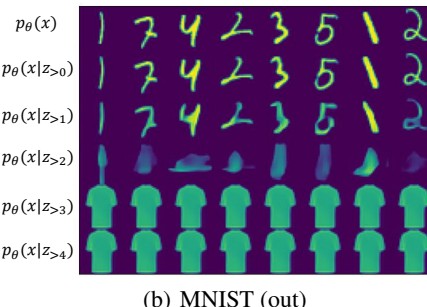

|(a) FashionMNIST (in)|(b) MNIST (out)|

Figure 2: Illustration of "*posterior collapse*" in a 5-layer hierarchical VAE trained on FashionMNIST (in) by visualizing its reconstructions conditioned on various $\boldsymbol{z}_{>k}$ for both in-distribution and OOD samples. The first row is the input $\boldsymbol{x}$, and the other rows are generated from the partial generative model $p_\theta(\boldsymbol{x}|\boldsymbol{z}_{>k})$ by taking $\boldsymbol{z}_{>k}$ drawn from $q_\phi(\boldsymbol{z}_{>k}|\boldsymbol{x})$ as input, where $k \in \{0, ..., 4\}$.

which is induced by the variational posterior $q_\phi(\boldsymbol{z}_{>k}|\boldsymbol{x})$. Then KL term in Eq. (3) can be rewritten as

$$
\mathbb{E}_{p(\boldsymbol{x})} \left[ \sum_{l=1}^{L} D_{\mathrm{KL}}(q_\phi(\boldsymbol{z}_l|\boldsymbol{z}_{l+1})||p_\theta(\boldsymbol{z}_l|\boldsymbol{z}_{l+1})) \right]
$$
$$
= \mathbb{E}_{p(\boldsymbol{x})} \left[ \sum_{l=1}^{k} D_{\mathrm{KL}}(q_\phi(\boldsymbol{z}_l|\boldsymbol{z}_{l+1})||p_\theta(\boldsymbol{z}_l|\boldsymbol{z}_{l+1})) \right] + \mathbb{E}_{p(\boldsymbol{x})} \left[ D_{\mathrm{KL}}(q_\phi(\boldsymbol{z}_{>k}|\boldsymbol{x})||p_\theta(\boldsymbol{z}_{>k})) \right] \quad (4)
$$
$$
= \mathbb{E}_{p(\boldsymbol{x})} \left[ \sum_{l=1}^{k} D_{\mathrm{KL}}(q_\phi(\boldsymbol{z}_l|\boldsymbol{z}_{l+1})||p_\theta(\boldsymbol{z}_l|\boldsymbol{z}_{l+1})) \right] + \mathcal{I}_q(\boldsymbol{x}, \boldsymbol{z}_{>k}) + D_{\mathrm{KL}}(q_\phi(\boldsymbol{z}_{>k})||p_\theta(\boldsymbol{z}_{>k})),
$$

where $q_\phi(\boldsymbol{z}_{>k}) = \mathbb{E}_{p(\boldsymbol{x})} \left[ q_\phi(\boldsymbol{z}_{>k}|\boldsymbol{x}) \right]$ and the detailed derivation can be found in Appendix A. By substituting Eq. (4) into Eq. (3), due to the non-negativity of mutual information and KL divergence, we can find that maximizing the ELBO is opposite to maximizing the mutual information $\mathcal{I}_q(\boldsymbol{x}, \boldsymbol{z}_{>k})$. When $\mathcal{I}_q(\boldsymbol{x}, \boldsymbol{z}_{>k})$ is minimized to zero, the variational posterior $q_\phi(\boldsymbol{z}_{>k}|\boldsymbol{x})$ will be independent of the data $\boldsymbol{x}$, which leads to the phenomenon of "*posterior collapse*".

### 3.2 Why "*posterior collapse*" limits Likelihood-Ratio for OOD Detection

OOD detection has become one of the most important applications of VAEs, which can be applied to filter OOD samples by setting a threshold on the score of log-likelihood term. However, some recent studies have shown that, in some cases, VAEs tend to achieve higher likelihoods on certain types of OOD samples [36], which makes the OOD detection rules based on likelihood threshold problematic. Recently, Havtorn et al. [27] reinterpreted this problematic behavior by providing evidence that the low-level features learned by VAEs generalize well across datasets and dominate the estimated likelihoods. Inspired by the alternative log-likelihood lower bound [29] that partly replaces the inference network with the generative model to highlight high-level features, formulated as

$$
\mathcal{L}_{\boldsymbol{x}}^{>k} = \log p(\boldsymbol{x}) - D_{\mathrm{KL}}(p_\theta(\boldsymbol{z}_{\le k}|\boldsymbol{z}_{>k})q_\phi(\boldsymbol{z}_{>k}|\boldsymbol{x})||p_\theta(\boldsymbol{z}|\boldsymbol{x})), \quad (5)
$$

Havtorn et al. [27] considered to subtract $\mathcal{L}_{\boldsymbol{x}}^{>k}$ from $\mathcal{L}_{\boldsymbol{x}}$ to cancel out the data distribution $\log p(\boldsymbol{x})$, resulting in a likelihood-ratio score for unsupervised OOD detection as

$$
\mathcal{LLR}^{>k} = D_{\mathrm{KL}}(p_\theta(\boldsymbol{z}_{\le k}|\boldsymbol{z}_{>k})q_\phi(\boldsymbol{z}_{>k}|\boldsymbol{x})||p_\theta(\boldsymbol{z}|\boldsymbol{x})) - D_{\mathrm{KL}}(q_\phi(\boldsymbol{z}_{\le k}|\boldsymbol{z}_{>k})q_\phi(\boldsymbol{z}_{>k}|\boldsymbol{x})||p_\theta(\boldsymbol{z}|\boldsymbol{x})), \quad (6)
$$

which discards the likelihood term to prevent the low-level features from dominating and measures divergence in the latent space to ensure that data should be in-distribution across all feature levels.

To intuitively understand the nature of success in the likelihood-ratio score and illustrate why alleviating *"posterior collapse"* in hierarchical VAEs can improve its performance on OOD detection, we provide an insightful analysis on $\mathcal{LLR}^{>k}$ in Eq. (6) by reformulating it as follows:

$$
\mathcal{LLR}^{>k} = \mathbb{E}_{q_\phi(\boldsymbol{z}_{>k}|\boldsymbol{x})} \left[ D_{\mathrm{KL}}(p_\theta(\boldsymbol{z}_{\le k}|\boldsymbol{z}_{>k})||p_\theta(\boldsymbol{z}_{\le k}|\boldsymbol{z}_{>k}, \boldsymbol{x})) - D_{\mathrm{KL}}(q_\phi(\boldsymbol{z}_{\le k}|\boldsymbol{z}_{>k})||p_\theta(\boldsymbol{z}_{\le k}|\boldsymbol{z}_{>k}, \boldsymbol{x}) \right]
$$
$$
\approx \mathbb{E}_{q_\phi(\boldsymbol{z}_{>k}|\boldsymbol{x})} \left[ D_{\mathrm{KL}}(p_\theta(\boldsymbol{z}_{\le k}|\boldsymbol{z}_{>k})||q_\phi(\boldsymbol{z}_{\le k}|\boldsymbol{z}_{>k})) \right],
$$
$$
(7)
$$

where the detailed derivations can be found in Appendix B. Eq. (7) shows that, when the inference network $q_\phi(\boldsymbol{z}_{\le k}|\boldsymbol{z}_{>k})$ can approximate the true posterior $p_\theta(\boldsymbol{z}_{\le k}|\boldsymbol{z}_{>k}, \boldsymbol{x})$ very well, thorough equivalent replacement, $\mathcal{LLR}^{>k}$ will approach the expected KL divergence between the prior $p_\theta(\boldsymbol{z}_{\le k}|\boldsymbol{z}_{>k})$

and variational posterior $q_\phi(\boldsymbol{z}_{\leq k}|\boldsymbol{z}_{>k})$. More specifically, conditioned on the expectation of $\boldsymbol{z}_{>k}$ drawn from its variational posterior $q_\phi(\boldsymbol{z}_{>k}|\boldsymbol{x})$, $\mathcal{LLR}^{>k}$ is developed to calculate the summation of $k$ KL divergence terms, measuring the distance between the lower-level variables $\boldsymbol{z}_{\leq k}$ drawn from the generative model $p_\theta(\boldsymbol{z}_{\leq k}|\boldsymbol{z}_{>k})$ and those from the variational inference network $q_\phi(\boldsymbol{z}_{\leq k}|\boldsymbol{z}_{>k})$.

The premise of applying $\mathcal{LLR}^{>k}$ to OOD detection is that, after training a hierarahical VAE on in-distribution samples, for each OOD sample, the latent variables $\boldsymbol{z}_{\leq k}$ generated from the generative model $p_\theta(\boldsymbol{z}_{\leq k}|\boldsymbol{z}_{>k})$ will be clearly distinct from those drawn from the variational inference network $q_\phi(\boldsymbol{z}_{\leq k}|\boldsymbol{z}_{>k})$. For ease of understanding the principle, after training a 5-layer hierarchical VAE on FashionMNIST, for each hidden layer, we exhibit the reconstructions of both in-distribution (FashionMNIST) and OOD (MNIST) data samples with the partial generative model $p_\theta(\boldsymbol{x}|\boldsymbol{z}_{>k})$ conditioned on the latent variables $\boldsymbol{z}_{>k}$ drawn from $q_\phi(\boldsymbol{z}_{>k}|\boldsymbol{x})$ as shown in Fig. 2. From the results, we can find that, when setting $k = 2$, the reconstructions of MNIST samples tend to reflect the high-level semantic structures learned from FashionMNIST, indicating that the generation mechanism of $p_\theta(\boldsymbol{x}|\boldsymbol{z}_{>2})$ seems to prevent accurate reconstruction of out-of-distribution data, which implies that a score function based on the distance between $p_\theta(\boldsymbol{z}_{\leq 2}|\boldsymbol{z}_{>2})$ and $q_\phi(\boldsymbol{z}_{\leq 2}|\boldsymbol{z}_{>2})$, like $\mathcal{LLR}^{>2}$ in Eq. (7), could be a promising metric for OOD detection.

However, when the phenomenon of "*posterior collapse*" occurs, the variational posterior $q_\phi(\boldsymbol{z}_{>k}|\boldsymbol{x})$ will be independent of the data $\boldsymbol{x}$, resulting in $q_\phi(\boldsymbol{z}_{>k}|\boldsymbol{x}) \approx p_\theta(\boldsymbol{z}_{>k})$, and the reconstructions of in-distribution and OOD samples, which are generated from the partial generative model $p_\theta(\boldsymbol{x}|\boldsymbol{z}_{>k})$, will be almost the same, such as the visualization examples shown in Fig. 2 by setting $k = 3$ or $k = 4$. In that case, for each in-distribution sample, the latent variables $\boldsymbol{z}_{\leq k}$ generated $p_\theta(\boldsymbol{z}_{\leq k}|\boldsymbol{z}_{>k})$ will be clearly distinct from those drawn from $q_\phi(\boldsymbol{z}_{\leq k}|\boldsymbol{z}_{>k})$, and similar conclusions can also be achieved by OOD samples, which will reduce the variance of $\mathcal{LLR}^{>k}$ scores between in-distribution and OOD samples and further bring troubles for OOD detection with $\mathcal{LLR}^{>k}$.

### 3.3 Informative Hierarchical VAE to Alleviate "*posterior collapse*"

Recall to the conflict between the ELBO objective and $\mathcal{I}_q(\boldsymbol{x}, \boldsymbol{z}_{>k})$ as discussed in Sec. 3.1, which causes "*posterior collapse*" in hierarchical VAEs, there could be two main approaches to alleviate this phenomenon, including: 1) downweight the KL term, like applying a warm-up scheme on it [37], which is the most common heuristic in practice but still cannot essentially address this issue; 2) enhance the connections between the observation and its multi-layer stochastic latent representations in the expected log-likelihood term, like modifying the generative process described by $p_\theta(\boldsymbol{x}|\boldsymbol{z})$ [29].

In this paper, focused on exploring the potential of alleviating "*posterior collapse*" with the second approach, we try to introduce skip-connection-liked structures into expected log-likelihood term to enhance the connections between the data $\boldsymbol{x}$ and the latent variables $\boldsymbol{z} = \boldsymbol{z}_1, ..., \boldsymbol{z}_L$. However, constrained by the layer-by-layer generation process of hierarchical VAE, there remains a great challenge to introduce physical skip connections into the generative model $p_\theta(\boldsymbol{x}|\boldsymbol{z})$, because arbitrarily adding or concatenating the stochastic hidden layers at different semantic levels will hurt the hierarchy of these multi-layer latent representations. We emphasize that the skip connections between the single stochastic layer and multiple deterministic layers [34] cannot be extended for hierarchical VAEs with multiple stochastic hidden layers, but our developed method below can be applied to any existing hierarchical VAE, which is one of the main contributions of this paper.

Generally speaking, moving beyond downweighting $\mathcal{I}_q(\boldsymbol{x}, \boldsymbol{z}_{>k})$ included in the KL term or modifying the structure of generative model $p_\theta(\boldsymbol{x}|\boldsymbol{z})$, our main idea is to upweight the mutual information between the data $\boldsymbol{x}$ and the higher-level variables $\boldsymbol{z}_{>k}$, which is denoted as

$$\mathcal{I}_p(\boldsymbol{x}, \boldsymbol{z}_{>k}) = -\mathcal{H}_p(\boldsymbol{x}|\boldsymbol{z}_{>k}) + \mathcal{H}_p(\boldsymbol{x}) = \mathbb{E}_{p(\boldsymbol{x})p_\theta(\boldsymbol{z}_{>k}|\boldsymbol{x})} \log p_\theta(\boldsymbol{x}|\boldsymbol{z}_{>k}) - \mathbb{E}_{p(\boldsymbol{x})} \log p(\boldsymbol{x}), \quad (8)$$

in the objective function of hierarchical VAEs. In Eq. (8), the first item can be approximated by $\mathcal{H}_{p,q}(\boldsymbol{x}|\boldsymbol{z}_{>k}) = \mathbb{E}_{p(\boldsymbol{x})q_\phi(\boldsymbol{z}_{>k}|\boldsymbol{x})} \log p_\theta(\boldsymbol{x}|\boldsymbol{z}_{>k})$ and $\mathcal{H}_p(\boldsymbol{x})$ is a constant, leading to the optimization direction of $\mathcal{I}_p(\boldsymbol{x}, \boldsymbol{z}_{>k})$ is consistent with $\mathcal{H}_{p,q}(\boldsymbol{x}|\boldsymbol{z}_{>k})$. Thus, targeted at directly maximizing multiple $\mathcal{H}_{p,q}(\boldsymbol{x}|\boldsymbol{z}_{>k})$, we develop an informative loss for training hierarchical VAEs, denoted as

$$\mathcal{L}^{in} = \mathbb{E}_{p(\boldsymbol{x})} \left[ \frac{1}{L} \sum_{k=0}^{L-1} \mathbb{E}_{q_\phi(\boldsymbol{z}_{>k}|\boldsymbol{x})} \left[ \log p_\theta(\boldsymbol{x}|\boldsymbol{z}_{>k}) \right] - \sum_{l=1}^{L} D_{\mathrm{KL}}(q_\phi(\boldsymbol{z}_l|\boldsymbol{z}_{l+1}) || p_\theta(\boldsymbol{z}_l|\boldsymbol{z}_{l+1})) \right] \quad (9)$$

where $p_\theta(\boldsymbol{x}|\boldsymbol{z}_{>k}) = \mathbb{E}_{p_\theta(\boldsymbol{z}_{\leq k}|\boldsymbol{z}_{>k})} \left[ p_\theta(\boldsymbol{x}|\boldsymbol{z}_{\leq k}) \right]$ describes a partial generative model to reconstruct the observation $\boldsymbol{x}$ taking $\boldsymbol{z}_{>k}$ drawn from the variational inference network $q_\phi(\boldsymbol{z}_{>k}|\boldsymbol{x})$ as input; the

weight $1/L$ before each expected term is introduced to keep numerical stability. The developed $\mathcal{L}^{in}$ not only inherits the terms of ELBO in Eq. (1), which helps preserve the original model properties of VAE, but also introduces virtual skip-connection-liked structures with partial generative models to enhance the connections between $\boldsymbol{x}$ and $\boldsymbol{z}_{>k}$, contributing to alleviating "*posterior collapse*".

To avoid directly calculating $p_\theta(\boldsymbol{x}|\boldsymbol{z}_{>k})$ in practice, inspired by [27, 29], the informative loss in Eq. (9) can be optimized by maximizing its lower bound $\hat{\mathcal{L}}^{in}$, expressed as

$$\mathcal{L}^{in} \geq \hat{\mathcal{L}}^{in} = \mathbb{E}_{p(\boldsymbol{x})}\left[\frac{1}{L}\sum_{k=0}^{L-1}\mathbb{E}_{p_\theta(\boldsymbol{z}_{\leq k}|\boldsymbol{z}_{>k})q_\phi(\boldsymbol{z}_{>k}|\boldsymbol{x})}\left[\log p_\theta(\boldsymbol{x}|\boldsymbol{z}_{\leq k})\right]\right]$$
$$- \mathbb{E}_{p(\boldsymbol{x})}\left[\sum_{l=1}^{L}D_{\mathrm{KL}}(q_\phi(\boldsymbol{z}_l|\boldsymbol{z}_{l+1})||p_\theta(\boldsymbol{z}_l|\boldsymbol{z}_{l+1}))\right], \tag{10}$$

where the expected log-likelihood term is the summation of $L$ components and each component denoted as $\mathcal{LL}^{>k} = \mathbb{E}_{p_\theta(\boldsymbol{z}_{\leq k}|\boldsymbol{z}_{>k})q_\phi(\boldsymbol{z}_{>k}|\boldsymbol{x})}\left[\log p_\theta(\boldsymbol{x}|\boldsymbol{z}_{\leq k})\right]$ can be obtained by replacing the inference network $q_\phi(\boldsymbol{z}_{\leq k}|\boldsymbol{z}_{>k})$ in the original log-likelihood term as described in Eq. (3) with the generative model $p_\theta(\boldsymbol{z}_{\leq k}|\boldsymbol{z}_{>k})$. Intuitively, for each expected log-likelihood term $\mathcal{LL}^{>k}$, after sampling the top variables $\boldsymbol{z}_{>k}$ from the variational posterior $q_\phi(\boldsymbol{z}_{>k}|\boldsymbol{x})$, these variables $\boldsymbol{z}_{>k}$ will be forced to reconstruct the observation $\boldsymbol{x}$ with the partial generative model $p_\theta(\boldsymbol{x}|\boldsymbol{z}_{>k})$, which builds the straightforward connections between $\boldsymbol{x}$ and $\boldsymbol{z}_{>k}$ to alleviate "*posterior collapse*".

We refer to the hierarchical VAE trained with the lower bound of informative loss in Eq. (10) as informative hierarchical VAE. We note that this method is applicable for hierarchical VAEs with either top-down or bottom-up inference network to explicitly utilize the generative hierarchy of the multi-layered stochastic variables during training, and can be flexibly extended in future works.

## 3.4  Adaptive Likelihood Ratio for OOD detection

Besides "*posterior collapse*", an inappropriate choice of $k$ in the likelihood-ratio score function, denoted as $\mathcal{LLR}^{>k}$ in Eq. (6), will also bring negative impact on the performance of applying hierarchical VAEs for OOD detection. Recall to the visualization examples in Fig. 2, when setting $k=0$ or $k=1$, the reconstruction quality of either in-distribution or OOD samples is surprisingly high, leading to the $\mathcal{LLR}^{>k}$ scores of both in-distribution and OOD samples are relatively small and further making it difficult to distinguish whether the data sample is OOD or not. Moving beyond cherry picking the hyperparameter $k$ on testing OOD samples, which is unreasonable for unsupervised OOD detection, we develop a novel adaptive likelihood-ratio score function $\mathcal{LLR}^{ada}$, described as

$$\mathcal{LLR}^{ada} = \sum_{k=0}^{L-1}\frac{\mathcal{R}(\boldsymbol{x},\boldsymbol{z}_{>k-1})}{\mathcal{R}(\boldsymbol{x},\boldsymbol{z}_{>k})}(\mathcal{LLR}^{>k} - \mathcal{LLR}^{>k-1}), \tag{11}$$

where $\mathcal{R}(\boldsymbol{x},\boldsymbol{z}_{>k})$ is designed to measure the relevance between the data sample $\boldsymbol{x}$ and its latent variables $\boldsymbol{z}_{>k}$ sampled from the variational posterior $q_\phi(\boldsymbol{z}_{>k}|\boldsymbol{x})$, specifically defining $\mathcal{LLR}^{>-1} := 0$ and $\mathcal{R}(\boldsymbol{x},\boldsymbol{z}_{>-1}) := \mathcal{R}(\boldsymbol{x},\boldsymbol{z}_{>0})$. More specifically, there are many choices for the definition of $\mathcal{R}(\boldsymbol{x},\boldsymbol{z}_{>k})$, but in the following experiments, we only use the log-likelihood score for brevity, by specifically defining $\mathcal{R}(\boldsymbol{x},\boldsymbol{z}_{>k}) := 1/\log p_\theta(\boldsymbol{x}|\boldsymbol{z}_{>k})$.

The intuition of designing $\mathcal{LLR}^{ada}$ is to move beyond the choose of $k$ but adaptively enhance the importance of some discriminative terms, like $\mathcal{LLR}^{>2}$, in the overall score function for OOD detection. With $\mathcal{R}(\boldsymbol{x},\boldsymbol{z}_{>k})$ to measure the relevance between $\boldsymbol{x}$ and $\boldsymbol{z}_{>k}$, we find that the adaptive weight $\frac{\mathcal{R}(\boldsymbol{x},\boldsymbol{z}_{>k-1})}{\mathcal{R}(\boldsymbol{x},\boldsymbol{z}_{>k})}$ will be relatively large when the data information drop rapidly at the current hidden layer, like $k=2$ in Fig. 2, which can be naturally applied as the importance weights to enlarge the gap between the metric scores of in-distribution and OOD samples. Compared to the previous score functions for OOD detection [27, 32], the developed $\mathcal{LLR}^{ada}$ in Eq. (11) owns less hyperparameters to be tuned, making its performance more stable on various benchmarks. More discussions about $\mathcal{LLR}^{ada}$ can be found in Appendix C.

Table 1: The comparisons of the 5-layer informative hierarchical VAE with $\mathcal{LLR}^{ada}$ and other OOD detection methods. The state-of-the-art results achieved by the methods of the category "Not ensembles" of "Unsupervised" have been bold.

| FashionMNIST(in)/MNIST(out) | | | | | | CIFAR10(in)/SVHN(out) | | | | | |
|---|---|---|---|---|---|---|---|---|---|---|---|
| Labels | | Prior | | Unsupervised | | Labels | | Prior | | Unsupervised | |
| Method | AUROC↑ | Method | AUROC↑ | Method | AUROC↑ | Method | AUROC↑ | Method | AUROC↑ | Method | AUROC↑ |
| CP [5] | 73.4 | LR(PC) [1] | 99.4 | *-Ensembles* | | MD [9] | 99.7 | LR(PC) [1] | 93.0 | *-Ensembles* | |
| CP(Ent) [5] | 74.6 | LR(BC) [1] | 45.5 | WAIC(5VAE) [24] | 76.6 | LMD [38] | 27.9 | LR(VAE) [1] | 26.5 | WAIC(5Glow) [24] | 99.0 |
| ODIN [8] | 75.2 | CP(OOD) [1] | 87.7 | WAIC(5PC) [24] | 22.1 | EN [11] | 98.9 | OE [28] | 98.4 | WAIC(5PC) [24] | 62.8 |
| VIB [6] | 94.1 | CP(Cal) [1] | 90.4 | *-Not Ensembles* | | iDE [14] | 95.7 | IC(Glow) [33] | 95.0 | *-Not Ensembles* | |
| MD(CNN) [9] | 94.2 | IC(Glow) [33] | 99.8 | LRe [32] | **98.8** | | | IC(PC++) [33] | 92.9 | LRe [32] | 87.5 |
| MD(DN) [9] | 98.6 | IC(PC++) [33] | 96.7 | HVK [27] | 98.4 | | | IC(HVAE) [33] | 83.3 | HVK [27] | 89.1 |
| DE [5] | 85.7 | | | $\mathcal{LLR}^{ada}$(Ours) | 98.0 | | | | | $\mathcal{LLR}^{ada}$(Ours) | **94.2** |

Table 2: The comparisons of the 3-layer informative hierarchical VAEs with various score functions and other unsupervised OOD detection methods.

| FashinMNIST(in)/MNIST(out) | | | | CIFAR10(in)/SVHN(out) | | | |
|---|---|---|---|---|---|---|---|
| Method | AUROC↑ | AUPRC↑ | FPR80↓ | Method | AUROC↑ | AUPRC↑ | FPR80↓ |
| WAIC(5PC) [24] | 22.1 | 40.1 | 91.1 | WAIC(5PC) [24] | 62.8 | 61.6 | 65.7 |
| HVK [27] | **98.4** | **98.4** | 1.3 | HVK [27] | 89.1 | 87.5 | 17.2 |
| *-Ours:* | | | | *-Ours:* | | | |
| $\mathcal{L}$ | 55.3 | 51.8 | 67.9 | $\mathcal{L}$ | 49.9 | 51.0 | 79.4 |
| $\mathcal{LLR}^{>1}$ | 97.5 | 97.0 | 2.8 | $\mathcal{LLR}^{>1}$ | 68.4 | 71.3 | 61.8 |
| $\mathcal{LLR}^{>2}$ | 97.4 | 97.7 | **1.2** | $\mathcal{LLR}^{>2}$ | **93.0** | **92.5** | **10.8** |
| $\mathcal{LLR}^{ada}$ | 97.0 | 97.6 | **0.9** | $\mathcal{LLR}^{ada}$ | 92.6 | 91.8 | 11.1 |

## 4 Experiments

### 4.1 Experimental setup

**Datasets:** Following [1, 27, 32], we compare our method with previous works on two dataset pairs, including: FashionMNIST [39] (in) / MNIST [40] (out) and CIFAR10 [41] (in) / SVHN [42] (out), where the suffix "in" and "out" denote the in-distribution dataset and OOD dataset, respectively. To better evaluate the generalization ability of these methods, we introduce additional OOD datasets: for FashionMNIST/MNIST pair, we add KMNIST [43], notMNIST [44], Omniglot [45] and SmallNORB [46] datasets; for CIFAR10/SVHN pair, we add CelebA [47], Places365 [48], Flower102 [49] and LFWPeople [50] datasets. More details about datasets can be found in Appendix D.

**Evaluation and Metrics:** We follow the evaluation procedure in Havtorn et al. [27], where all methods are trained on the training split of the in-distribution dataset, and their OOD detection performance is evaluated on both the testing split of the in-distribution dataset and OOD dataset. Following previous works' evaluation approaches [5, 6, 28], we adopt two popular threshold-independent evaluation metrics, including Area Under the Receiver Operator Characteristic (AUROC↑) and Area Under the Precision Recall Curve (AUPRC↑), and another metric False Positive Rate at 80% true positive rate (FPR80↓), where the arrow indicates the direction of improvement.

**Baselines:** The comparisons in our experiments mainly include two aspects: **i**) the comparisons with previous OOD detection methods to see whether our method can achieve competitive performance; **ii**) the comparisons with several hierarchical VAEs to see whether the new training objective of our method can lead to better performance. For the comparisons in **i**, the baselines can be divided into three categories: "**Labels**": methods using in-distribution data labels [5, 6, 8, 9, 38, 51]; "**Prior**": methods using the prior knowledge collected from OOD data [1, 28, 33]; and "**Unsupervised**": methods without any OOD-specific assumptions [24, 27, 32]. For the comparisons in **ii**, we compare our method with a normal bottom-up inference hierarchical VAE (HVAE) [20], which is also the backbone of our method, and its two major variants: a top-down inference hierarchical VAE named Ladder VAE (LVAE) [30] and a bidirectional inference hierarchical VAE (BIVA) [29]. More details of these baselines and the categories they belong to can be found in Appendix E.

**Implementation Details:** For the comparisons on FashionMNIST(in)/MNIST(out), we set the network structure of hierarchical VAEs as [16, 8, 4] and [32, 24, 16, 8, 4] from shallow to deep, respectively. For CIFAR10(in)/SVHN(out), we set the network structure as [128, 64, 32] and [128, 64, 32, 28, 24], respectively. For optimization, we adopt the same Adam optimizer [52] with a learning rate of 3e-4. We train all models in comparison by setting the batch size as 128 and the max epoch as

Table 3: The comparisons of the OOD detection performance of various 3-layer hierarchical VAEs with the same $\mathcal{LLR}^{ada}$ score function. "M1" refers to the metric AUROC↑, "M2" refers to the metric AUPRC↑, and "M3" refers to the metric "FPR80↓".

| OOD | Trained on FashionMNIST. | | | | | | | | | | | | Trained on CIFAR10. | | | | | | | | | | | |
| --- | --- | --- | --- | --- | --- | --- | --- | --- | --- | --- | --- | --- | --- | --- | --- | --- | --- | --- | --- | --- | --- | --- | --- | --- |
| | KMNIST | | | Omniglot | | | notMNIST | | | SmallNORB | | | CelebA | | | Places365 | | | Flower102 | | | LFWPeople | | |
| Model | M1 | M2 | M3 | M1 | M2 | M3 | M1 | M2 | M3 | M1 | M2 | M3 | M1 | M2 | M3 | M1 | M2 | M3 | M1 | M2 | M3 | M1 | M2 | M3 |
| HVAE [27] | 86.4 | 89.7 | 29.0 | 99.8 | 99.9 | **0.00** | 85.3 | 88.1 | 29.2 | **100** | **100** | **0.00** | 39.8 | 44.7 | 90.0 | 40.1 | 46.6 | 94.0 | 45.2 | 51.7 | 92.0 | 42.5 | 48.2 | 92.5 |
| LVAE [30] | 85.9 | 87.7 | 24.8 | 93.1 | 96.0 | 0.5 | 94.0 | 93.6 | 6.0 | 97.3 | 97.7 | 0.8 | 53.1 | 54.2 | 80.5 | 56.2 | 53.7 | 74.4 | 56.5 | 52.3 | 70.9 | 63.0 | 65.8 | 61.6 |
| BIVA [29] | 86.5 | 87.0 | 27.0 | **100** | **100** | **0.00** | 96.4 | 97.0 | 2.4 | 98.7 | 98.6 | 1.9 | 70.5 | 67.8 | 53.2 | 60.1 | **63.0** | 74.6 | 61.9 | 69.2 | 84.4 | 75.2 | 74.0 | 44.6 |
| Ours | **95.0** | **95.1** | **7.1** | **100** | **100** | **0.00** | **99.7** | **99.8** | **0.00** | **100** | **100** | 0.1 | **72.1** | **70.5** | **49.0** | **63.3** | 62.1 | **62.6** | **63.4** | **70.1** | **71.2** | **83.0** | **83.4** | **29.0** |

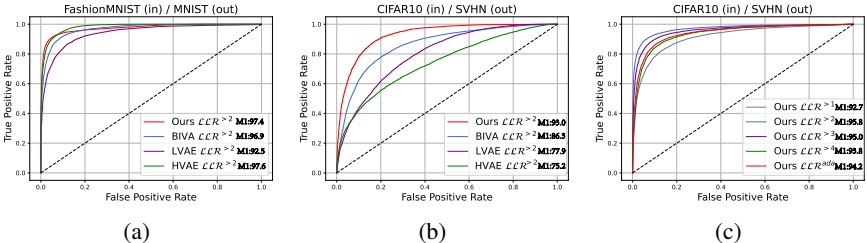

(a)  (b)  (c)

Figure 3: Plots of the ROC curves on the OOD detection performance of various hierarchical VAEs. (a)∼(b): ROC curves on FashionMNIST (in) / MNIST (out) and CIFAR10 (in) / SVHN (out) with the same score function $\mathcal{LLR}^{>2}$. (c): ROC curves on CIFAR10 (in) / SVHN (out) with various score functions ($\mathcal{LLR}^{>k}$ and $\mathcal{LLR}^{ada}$). "M1" denotes for the AUROC value.

1000. All experiments are performed on a PC with an NVIDIA RTX 3090 GPU and the our code is implemented with PyTorch [53]. More implementation details can be found in Appendix F.

## 4.2 Quantitative Comparisons

**Overall Comparisons:** Following the experimental settings in Sec. 4.1, we exhibit the experimental results in in Tab. 1. From the results, we can find that our method with the $\mathcal{LLR}^{ada}$ score function is comparable with those non-ensemble completely unsupervised methods in FashionMNIST/MNIST, and significantly outperform them in CIFAR10/SVHN. We emphasize that, without utilizing the labels of in-distribution samples [5, 6, 8, 9, 38, 51] or the prior knowledge collected from OOD samples [1, 28, 33], our method can still achieve competitive performance with these methods.

**Effectiveness of $\mathcal{LLR}^{ada}$:** Focused on the comparison between $\mathcal{LLR}^{>k}$ and $\mathcal{LLR}^{ada}$ exhibited in Tab. 2 and Fig. 3(c), when the performance of $\mathcal{LLR}^{>k}$ is sensitive to the selection of $k$, we can find that the performance of $\mathcal{LLR}^{ada}$ can approach the best performance achieved by $\mathcal{LLR}^{>k}$ with the optimal $k$, as shown in the right part of Tab. 2 (CIFAR/SVHN). Furthermore, when the performance of $\mathcal{LLR}^{>k}$ is stable, the developed $\mathcal{LLR}^{ada}$ can still achieve comparable OOD detection performance, as shown in the left part of Tab. 2 (FashionMNIST/MNIST) and Fig. 3(c). The experimental results above demonstrate the adaptability of our developed $\mathcal{LLR}^{ada}$.

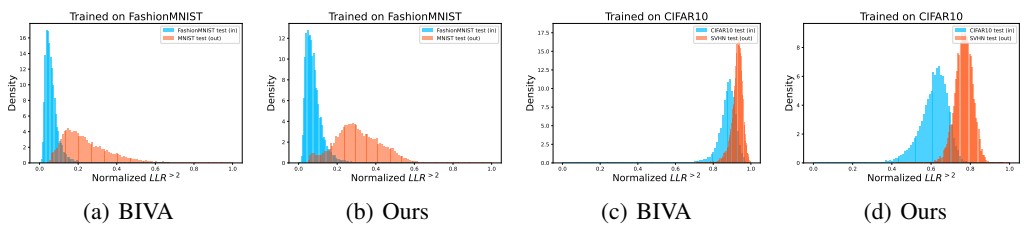

(a) BIVA  (b) Ours  (c) BIVA  (d) Ours

Figure 4: Empirical densities under the score function $\mathcal{LLR}^{>2}$ on FashionMNIST (in)/MNIST (out) and CIFAR10 (in)/SVHN (out) dataset pairs.

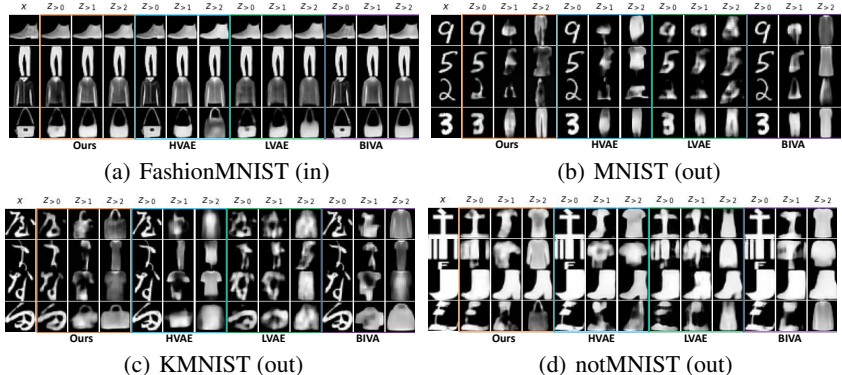

(a) FashionMNIST (in)    (b) MNIST (out)

(c) KMNIST (out)    (d) notMNIST (out)

Figure 5: Comparisons of reconstructions with 3-layer hierarchical VAEs trained on FashionMNIST, where the leftmost column in each subfigure is the input $x$ and the column noted with $z_{>k}$ means the generation from the partial generative model $p_\theta(x|z_{>k})$ with $k \in \{0, 1, 2\}$.

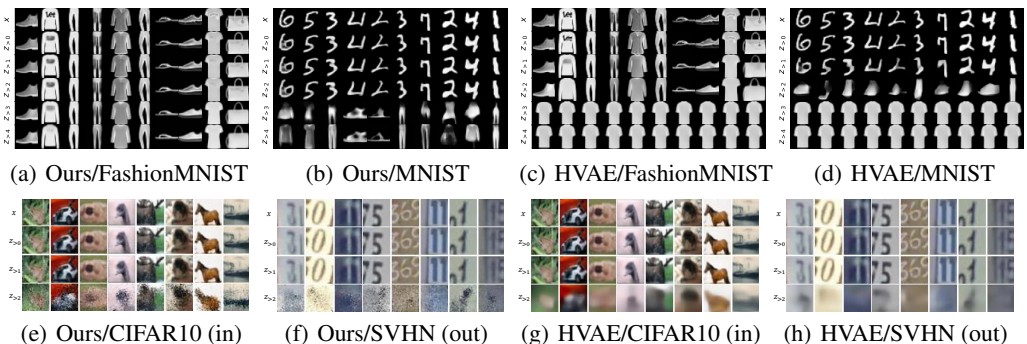

(a) Ours/FashionMNIST    (b) Ours/MNIST    (c) HVAE/FashionMNIST    (d) HVAE/MNIST

(e) Ours/CIFAR10 (in)    (f) Ours/SVHN (out)    (g) HVAE/CIFAR10 (in)    (h) HVAE/SVHN (out)

Figure 6: Comparisons of the degree of "*posterior collapse*" in 5-layer hierarchical VAEs. For each subfigure, the 1st row is the input image $x$ and the $i$th row is generated from $p_\theta(x|z_{>k})$.

**Effectiveness of Informative Hierarchical VAE:** Taking the same $\mathcal{LLR}^{ada}$ as the score function, we compare the performance of informative hierarchical VAE with other hierarchical VAEs in Tab. 3. From the results, we can find that our model outperforms others, indicating the effectiveness of alleviating "*posterior collapse*" with informative hierarchical VAE for OOD detection.

To evaluate the degree of "*posterior collapse*", we use the ROC curves in Fig. 3 and the empirical densities in Fig. 4 to compare the performance of $\mathcal{LLR}^{>2}$ scores based on the top-level latent variables $z_2$ of these VAEs. The ROC results in Fig. 3 demonstrate the superiority of the informative hierarchical VAE, confirming that our model can provide more expressive higher-level latent representations for OOD detection. The empirical densities in Fig. 4 show that $z_2$ learned by our model on the CIFAR10 (in)/SVHN (out) pairs owns better separability than those learned by BIVA.

## 4.3 Qualitative Analysis

**Meaningful Semantic Space Learned by Informative Hierarchical VAE:** Following the same procedure as Fig. 2, we visualize more reconstructed samples on various benchmarks in Fig. 5. From the visualized results, for both in-distribution and OOD samples, we can find that the quality of the reconstructions generated from $p_\theta(x|z_{>0})$ is surprisingly high, indicating that these reconstructions are almost dominated by the low-level features, which potentially explains the previous problematic phenomenon that these methods based on single-layer likelihood will assign higher likelihood scores for OOD samples and fail on OOD detection. Focused on the reconstructions generated by $p_\theta(x|z_{>2})$, we can find that the developed informative hierarchical VAE can provide more realistic and clear reconstructed samples for both in-distribution and OOD inputs, indicating that our model can learn a more meaningful high-level latent semantic space than other models. Furthermore, based on providing higher-quality reconstruction for OOD samples, the gap between the metric scores of

in-distribution and OOD samples in informative hierarchical VAE tend to be larger than other models, leading to a better OOD performance as shown in Tab. 3.

**Alleviating "*Posterior Collapse*" with Informative Hierarchical VAE:** As discussed in Sec. 3.1, "*posterior collapse*" will cause the higher-level latent variables to become uninformative. Considering the developed informative hierarchical VAE shares the same network structure with the basic HVAE, in this part, we focus on evaluating whether our model can alleviate "*posterior collapse*" in higher layers when the network depth becomes deeper. As shown in Fig. 6, for both in-distribution data and OOD data samples, the overall quality of the reconstructions generated by our model is significantly higher than those generated by the basic HVAE. Specifically, for FashionMNIST (in)/MNIST (out), the reconstructions generated by $p_\theta(\boldsymbol{x}|\boldsymbol{z}_{>3})$ and $p_\theta(\boldsymbol{x}|\boldsymbol{z}_{>4})$ of HVAE are almost the same, indicating that its posterior described by $q_\phi(\boldsymbol{z}_{>3}|\boldsymbol{x})$ or $q_\phi(\boldsymbol{z}_{>4}|\boldsymbol{x})$ tend to collapse to a prior distribution about T-shirts. On the contrary, the reconstructions generated by the our model are still realistic, where the reconstructions of MINST (out) samples generated by $q_\phi(\boldsymbol{z}_{>3}|\boldsymbol{x})$ or $q_\phi(\boldsymbol{z}_{>4}|\boldsymbol{x})$ preserve the semantic structural information learned form FashionMNIST (in), explaining the underlying reason why our model can achieve better OOD detection performance. Similar conclusions can be achieved by the experimental results on CIFAR10 (in)/SVHN (out) as shown in Fig. 6.

## 5 Conclusion

In this paper, after presenting a thorough analysis of "*posterior collapse*", we develop a novel informative hierarchical VAE to extract more expressive hierarchical latent representations by alleviating "*posterior collapse*". Then we theoretically explain why "*posterior collapse*" will limit the performance of existing hierarchical VAEs, and develop a novel Adaptive Likelihood Ratio score function for unsupervised OOD detection. Experiments demonstrate the effectiveness of our method, whose main thought can be borrowed other hierarchical VAEs to improving their performance on downstream tasks relied on the hierarchy of latent representations.

## 6 Acknowledgment

This research is supported by the National Research Foundation, Singapore under its Industry Alignment Fund – Pre-positioning (IAF-PP) Funding Initiative and Competitive Research Programme (Grant No. NRF-CRP23-2019-0006). Any opinions, findings and conclusions or recommendations expressed in this material are those of the author(s) and do not reflect the views of National Research Foundation, Singapore. Additionally, Tongliang Liu is partially supported by Australian Research Council Projects DP180103424, DE-190101473, IC-190100031, DP-220102121, and FT-220100318. Xiaobo Xia is supported by Australian Research Council Projects DE-190101473.

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
