# OpenReview forum: "Out-of-Distribution Detection with An Adaptive Likelihood Ratio on Informative Hierarchical VAE"
_NeurIPS.cc/2022/Conference — NeurIPS 2022 Accept_

### Official Review · Reviewer_9EQd · 2022-07-01

**Rating:** 7
**Confidence:** 4
**Soundness:** 3 good
**Presentation:** 3 good
**Contribution:** 3 good

**Summary:**

The paper aims to detect OOD samples (unsupervisedly) with hierarchical VAEs. The main idea is based on the likelihood ratio in [1], with some interesting modifications. Firstly, the paper demonstrates that alleviating posterior collapse in hierarchical VAEs can help the performance of OOD detection using likelihood ratio. Then the paper proposes a modified training objective that upweights the mutual information between data $x$ and higher level latent variables, which enforces higher level latent variables to contain information about $x$ and hence prevents higher level latent variables to collapse to prior. The paper also proposes a new score that eliminates the need to tune the hyper-parameter $k$ in previous likelihood ratio method.

[1] Hierarchical VAEs Know What They Don't Know

**Questions:**

1. How does the modification in training objective affect the performance of VAE, e.g., reconstruction, test likelihood, sample quality? We want the VAE to be versatile, not just a tool to detect OOD samples.

2. If we alleviate the posterior collapse with simpler approaches, such as simply downweighting the KL term or using some warm-up scheme, can we obtain hierarchical VAEs that do better in OOD detection?

3. In section 3.4, as well as appendix C, if we can use the ration of $R$, where $R$ is the partial generative model's log likelihood, why don't we just automatically pick k with the largest $R$-ratio?

4. In Figure 2, I don't quite understand why the partial reconstructions for $p_{\theta}(x|z_{>3})$ and $p_{\theta}(x|z_{>4})$ have absolutely no variation. Assuming $z_{>3}$ has been collapsed into prior, then $q(z_{>3}|x) \approx p(z_{>3})$. As a result,  sampling $z_{>3}$ from posterior $q(z_{>3}|x)$ and $z_{<3}$ from prior $p(z_{<3}|z_{>3})$ (which is what a partial generative model does) is essentially sampling the whole $z$ from prior. Then the resulting reconstruction should just be some generated samples from the VAE, which should be diverse. But here they are the same. Does the posterior collapse to a single point?

**Limitations:**

The authors have adequately addressed the limitations and potential negative societal impact of their work

**Strengths And Weaknesses:**

Strengths:

1.One thing I particularly like about the paper is that the proposed method is based on solid motivations. The paper clearly demonstrate the impact of posterior collapse to OOD detection in hierarchical VAEs, and then proposes method to alleviate posterior collapse. This makes the  main structure of the paper easy to follow.

2. The proposed method is simple yet effective. Upweighting the MI between $x$ and $z$ can certainly alleviate posterior collapse, but the magic only happens when it is combined with the analysis on the relationship between posterior collapse and OOD detection.

3. Comprehensive and strong experimental results.

Limitations:

1. Since the method is based on previous likelihood ratio idea, in order to make the paper self-contained, more details of the previous method should be given. In particular, section 3.2 should give a more detailed introduction to [1]. The motivation and intuitive explanation should be provided.

2. Section 3.4 is not very clear to me. Some explanations in the appendix should be moved to the main text.

[1] Hierarchical VAEs Know What They Don't Know

---

> ### Author Response · Authors · 2022-08-02
> **Response to Reviewer 9EQd**
>
> Thanks for your interest in our work!
>
> **For Limitations 1**
>
> Thanks for your suggestions. We have given an intuitive explanation of [1] in our revision (Section 3.2), and left the derivation details to Appendix B.
>
> **For Limitations 2**
>
> Thanks for your suggestions. We have revised Section 3.4 and give a brief explanation here.
>
> Firstly, we need to point out that, for the likelihood-ratio score function $\mathcal{LLR}^{>k}$ [19], cherry picking the hyperparameter $k$ on testing OOD samples is unreasonable for unsupervised OOD detection, but an inappropriate choice of $k$ will bring negative impact on the performance.
> Thus, the intuition of designing $\mathcal{LLR}^{ada}$ is to move beyond the choose of $k$ but adaptively enhance the importance of some discriminative terms, like $\mathcal{LLR}^{>2}$, in the overall score function for OOD detection.
> With $\mathcal{R}(x, z_\{>k\})$ to measure the relevance between $x$ and $z_\{>k\}$, we find that the adaptive weight $\frac{\mathcal{R}(x, z_\{>k-1\})}{\mathcal{R}(x, z_\{>k\})}$ will be relatively large when the data information drop rapidly at the current hidden layer, like $k=2$, which naturally meets our requirements for designing $\mathcal{LLR}^{ada}$.
>
> Through combining $\frac{\mathcal{R}(x, z_\{>k-1\})}{\mathcal{R}(x, z_\{>k\})}$ and $\mathcal{LLR}^{>k}$, there could be several ways to design the final score function. The reason why we choose $\mathcal{LLR}^{ada}$ in Eq. (10) is that it can numerically omit some terms that occur ``posterior collapse'', as discussed in item 4 to review jV8d.
>
> **For Questions 1**
>
> Thanks for your suggestions. We absolutely agree that any VAE-based extension should inherit the original model properties of VAE. In such consideration, we have included more VAE-related experiments in Appendix N and O, including
> 1) measure of reconstruction quality with partial generative models $p_\theta(\mathbf{x}|\mathbf{z}_{>k})$ (Table 15 in Appendix O);
> 2) t-sne visualization of hierarchical latent representations (Fig. 7 in Appendix N);
> 3) {\color{cyan}visualization of the data samples generated from the prior distribution} (Fig. 9 in Appendix O).
> From these results, we can find that our method can perverse the versatility of VAE.
>
> **For Questions 2**
>
> Thanks. We have provided additional comparisons of various methods to alleviate "posterior collapse'' of HVAEs in Table 6 and Table 7 of Appendix J.
> As shown in the results, with the same score function for OOD detection, HVAEs trained with the warm-up scheme can outperform the vanilla VAE without any modification on ELBO.
>
>
> **For Questions 3**
>
> Thanks for your awesome suggestion. As discussed in Q.2, there could be several ways to design the final score function through combining $\frac{\mathcal{R}(\mathbf{x}, \mathbf{z}_\{>k-1\})}{\mathcal{R}(\mathbf{x}, \mathbf{z}_\{>k\})}$ and $\mathcal{LLR}^\{>k\}$, where automatically picking $k$ with the largest $R$-ratio could also  be an interesting and effective way.
> We note that we have included the method you suggested as a score function baseline, termed $\mathcal{LLR}^{opt_k}$, in our experiments, as shown in Table 10, Table 11, and Table 12 of Appendix.
> From the results, we can find that $\mathcal{LLR}^{ada}$ can stably outperform $\mathcal{LLR}^{opt_k}$. The potential reason could be that the optimal choices of $k$ for data samples are quite different, and the non-unified scales of scoring functions will cause confusion for OOD detection (the scales of $\mathcal{LLR}^{>1}$ and $\mathcal{LLR}^{>2}$ are different).
> Thus, $\mathcal{LLR}^{ada}$ could be a more stable and soft choice when compared to $\mathcal{LLR}^{opt_k}$.
>
> **For Questions 4**
>
> We appreciate it so much for your careful reviewing our paper!
> The reason is that we only exhibit the reconstructed samples with the highest probability in Fig. 2.
> Specifically, for each hidden layer $l$, we deterministically estimate $z_l$ with the mean vector of Gaussian-distributed $p_{\theta}(z_l|z_{>l})$ without noise sampling, resulting in that the data samples generated from HAVE with $p_\theta(x|z_{>3})$ and $p_\theta(x|z_{>4})$ are exactly same when the estimated posterior $q_\phi(z_4|z_5, x)$ collapses to its prior $p_\theta(z_4|z_5)$. As shown in Table 14 of Appendix N, the KL-divergence scores of the 4-th and 5-th hidden layers are almost close to zero, which indicates the appearing of `"posterior collapse''.
>
>
> To intuitively demonstrate that the posterior does not collapse to a single point,  we visualize the data samples generated from $p_\theta(x|z_{>k})$ by taking the latent variables $z_k$ sampled from the posterior $q_\phi(z_{k}|z_{>k}, x)$ as input, where $x$ is a fixed data point. As shown in Fig. 8 of Appendix~N, the diversity of the generated samples demonstrate that the posterior $q_\phi(z_{k}|z_{>k}, x)$ collapses to its prior distribution $p_\theta(z_k|z_{>k})$ rather than a single point.

---

> > ### Comment · Reviewer_9EQd · 2022-08-04
> > **Thank you for the response**
> >
> > I have read the response and revised paper, and my concern has been resolved. I would like to increase my score to 7.

---

### Official Review · Reviewer_jV8d · 2022-07-12

**Rating:** 7
**Confidence:** 3
**Soundness:** 3 good
**Presentation:** 3 good
**Contribution:** 3 good

**Summary:**

The work studies OOD in hierarchical VAEs (HVAE). It connects the earlier observations that HVAEs may yield higher likelihood for OOD samples to the so-called “posterior collapse”, where higher-level latent variables degenerate to the (conditional) prior and hence becomes uninformative w.r.t. the input sample. To alleviate this issue, the work promotes increased mutual information between the input and the higher-level latents. It further develops an OOD score based on a weighted difference of the log-likelihood ratio between subsequent slices in the hierarchy of the latent variables.
Experiments demonstrate improved OOD accuracy on two benchmarks. The proposed OOD criterion is shown to be more stable than the layer-specific scores (which implicitly require the index of the layer as a hyperparamter).

**A post-rebuttal note.**
I thank the authors for their elaboration and I appreciate the effort. I increase my score, since my main concerns have been resolved. Nevertheless, I encourage the authors to improve clarity in the main text, as well as to include the computational considerations entailed by the approach (as provided in the response below) in the final revision.

**Questions:**

- What is the OOD accuracy with the vanilla VAE loss (i.e. without the informative loss), but with the adaptive criterion?
- Testing the approach on MNIST and CIFAR is great, but what about more natural images (e.g. ImageNet)?
- How does the computational footprint change specifically w.r.t. the baseline?


**Limitations:**

The limitations in Sec. H are a bit unspecific ("additional computational burden"). The broader impact in Sec. I is perhaps a bit more technical than it should be.

**Strengths And Weaknesses:**

**Pros:**
- the narrative that OOD accuracy in hierarchical VAEs is connected to the posterior collapse issue is compelling and interesting to read.
- the work appears technically sound and the arguments are appropriately formalised.
- the proposed OOD score does not have any hyperparameters, but nevertheless appears competitive w.r.t. parametric alternatives.
- the empirical results are strong and support the main claims.


**Cons:**

While the OOD is the focus of this work, the approach leads to increased in-domain likelihood overall, in some cases rather substantially so (c.f. Tab. 4 in Appendix);

Both the proposed training and the adaptive OOD score may be a suboptimal choice. For example, on FashionMNIST->MNIST the likelihood regret approach has a somewhat superior OOD accuracy.

Comparisons could have been a bit more extensive (only 2 training sets). Tab. 3 and 4 in the appendix provide more results, but those are not compared to previous non-VAE and more recent works (e.g. [A])

The idea behind the proposed score seems to be mainly intuition-based and lacks theoretical backing. The explanation in Appendix C does not provide further theoretical insights.

[A] Revisiting Flow Generative Models for Group-wise Out-of-Distribution Detection

**Typos:**
- l. 70 “inference”
- l. 100 “PixelCNN”
- l. 206,213 “ration”
- l. 310 “still” redundant
- l. 311 “preserve”

**Other comments:**
- Tab. 2 left should probably read “FashionMNIST(in)/MNIST(out)”
- The exposition could be improved (e.g. l. 95 “at the cost of bringing heavy burdens”). It is not always clear in notation, if it is a definition or an implied equality (e.g. l. 120,188).
- The acronyms used in Tab. 1 are not all self-explanatory, nor is it very clear which previous works specifically they come from. Please make it clearer in the revision.

---

> ### Author Response · Authors · 2022-08-02
> **Response to Reviewer jV8d (1/3)**
>
> Thanks for your interest in our work!
>
> **1. For "While the OOD is the focus of this work, the approach leads to increased in-domain likelihood overall, in some cases rather substantially so. (c.f. Tab. 4 in Appendix)"**
>
> Thanks for your careful review.
> Firstly, we need to highlight that the metric $L_x$ in Table. 4 is the ELBO ($\log p(\mathbf{x})$ rather than the reconstruction log-likelihood ($\log p(\mathbf{x}|\mathbf{z})$). Then, what we want to highlight in Table 4 is that our method can lessen the gap between $L_x$ and $L_x^{>4}$ (a smaller $LLR^{>4}$) for in-distribution testing data samples, illustrating why our method can outperform the other baselines.
>
> To better understand why our method will lead to an increased in-domain $L_x$, which consists of an expected likelihood term and several KL divergence terms, we provide additional comparisons of layer-wise log-likelihood and KL divergence between HVAE and our method, as shown in Table 15 and Table 14 of Appendix respectively.
> As the log-likelihood results shown in Table 15, our method can achieve comparable performance with HVAE at the first-layer likelihood $p_\theta(\mathbf{x}|\mathbf{z}_{>0})$, and significantly outperform it at higher-level likelihoods $p_\theta(\mathbf{x}|\mathbf{z}_\{>k\})$ for $k>0$. From the KL-divergence results shown in Table 14, we can find that the KL-divergence scores of our method will be larger than those of HVAE at higher layers, indicating that our method can effectively alleviate ``posterior collapse'' at higher layers.
> Thus, a comparable first-layer likelihood $p_\theta{(\mathbf{x}|\mathbf{z}_\{>0\})}$ and a larger summation of KL-divergence terms together lead to a decreased in-domain $L_x$ for our method, which leads to a higher Average bits per dim.
>
> To make a comprehensive comparison, we also measure the reconstruction quality with partial generative models $p_\theta(\mathbf{x}|\mathbf{z}_\{>k\})$ and visualize the data samples generated from the prior distribution in Appendix O.
>
> **2. For "Both the proposed training and the adaptive OOD score may be a suboptimal choice. For example, on FashionMNIST->MNIST the likelihood regret approach has a somewhat superior OOD accuracy."**
>
> For the training scheme in informative HVAE, it has been proven to be effective to alleviate ``posterior collaspe'' and further improve the performance of unsupervised OOD detection in all our experiments.
> For $LLR^{ada}$,  in some cases, we admit that it will be a suboptimal choice compared to Likelihood Regret (only 0.8\% smaller in AUROC when detecting MNIST as OOD). However, we need to point out that the calculation of Likelihood Regret is extremely time-consuming. For **each** testing sample, after fixing the parameters in the encoder network of VAE obtained by pretraining, it requires the model to iteratively
> finetune the parameters of the decoder network blue **only for one data sample** until convergence to calculate Likelihood Regret score for it, which can hardly achieve fast in out-of-sample prediction and be applied in real-world applications (not all machines support the finetune). On the contrary, the calculation of $LLR^{ada}$ is straightforward and is more suitable for real-time prediction.
>
> **3. For "Comparisons could have been a bit more extensive (only 2 training sets). Not compared to previous non-VAE and more recent works (e.g. [A])"**
>
> Thanks for your recommending the paper [A].
> To make a comprehensive comparison with non-VAE deep likelihood-based }models on unsupervised OOD detection, we provided additional experimental results in Table 8 and Table 9 of Appendix K, including Flow+Group [A], Glow [1], and PixcelCNN++ [2].
> Specifically, Flow+Group [A] is an SOTA flow-based group OOD detection method to justify whether a batch of samples \{${x_1, x_2, ..., x_n}$\} $(n>1)$ is an OOD batch, rather than a sample-level OOD detection method ($n=1$}).
> Luckily, the authors have extended their method [A] via data augmentation to a sample-level OOD detection situation, i.e., $n=1$, whose setting is the same as us, and therefore we directly cite their OOD detection results reported in Appendix F;
> For Glow [1], which thoroughly shows that flow-based models tend to assign higher likelihood scores to OOD samples, we report the results with their released code for OOD detection;
> For PixcelCNN++ [2], which proposes to use an auto-regressive model for OOD detection with the help of additional OOD datasets, like NotMNIST dataset, we report the results of PixcelCNN++ under our purely unsupervised setting (no additional datasets).
>
> From the results shown in Table 8 and Tabl 9 of Appendix K, we can find that Flow+Group and our method can significantly outperform the other non-VAE methods, while our method is still better than Flow+Group [A].
>
> [1] Nalisnick et al. "Do deep generative models know what they don't know?".
>
> [2] Ren et al. "Likelihood Ratios for Out-of-Distribution Detection".

---

> > ### Author Response · Authors · 2022-08-02
> > **Response to Reviewer jV8d (2/3)**
> >
> > **4. For "The idea behind the proposed score seems to be mainly intuition-based and lacks theoretical backing. The explanation in Appendix C does not provide further theoretical insights."**
> >
> > Thanks for your suggestions! The design of the adaptive score function is mainly inspired by the insight that the adaptive weight $\frac{R(x, z_\{>k-1\})}{R(x, z_\{>k\})}$ will be relatively large when the data information drop rapidly, and can be used to adaptively enhance the importance of some discriminative terms, like $LLR^\{>2\}$, in the overall score function for OOD detection. Compared to $LLR^\{>k\}$'s unreasonably cherry picking $k$ on the whole testing set, the developed $LLR^\{ada\}$ does move beyond the choice of $k$ and still achieve competitive OOD detection performance in an unsupervised manner.
> >
> > For theoretical analysis, we note that the  $LLR^\{ada\}$ in Eq. (10) can be rewritten as
> > $$\\begin{aligned}
> > LL{R^{ada}} = \\frac{{R(x,{z\_{ &gt; L - 2}})}}{{R(x,{z\_{ &gt; L - 1}})}}LL{R^{ &gt; L - 1}} + \\sum\\nolimits\_{k = 0}^{L - 2} {(\\frac{{R(x,{z\_{ &gt; k}})}}{{R(x,{z\_{ &gt; k + 1}})}} - \\frac{{R(x,{z\_{ &gt; k - 1}})}}{{R(x,{z\_{ &gt; k}})}})LL{R^{ &gt; k}}}  - LL{R^{ &gt;- 1}}.\\end{aligned}$$
> >
> > Recall to the visualization exhibited in Fig. 2, the weight score of $LLR^{>k}$ will be close to zero when ``posterior collapse'' occurs, like $k=3$, because no information decay will cause $\\frac{{R(x,{z\_{ &gt; k}})}}{{R(x,{z\_{ &gt; k + 1}})}} = \\frac{{R(x,{z\_{ &gt; k - 1}})}}{{R(x,{z\_{ &gt; k}})}} = 1$;
> > when the data information suddenly drop rapidly, like $k=2$, the weight score will be relatively large, leading to $\frac{{R(x,{z_{ > k}})}}{{R(x,{z_{ > k + 1}})}} 	\gg \frac{{R(x,{z_{ > k - 1}})}}{{R(x,{z_{ > k}})}}$; on the contrary, if the data information drop slowly, the weight score will be relatively small, because  $\frac{{R(x,{z_{ > k}})}}{{R(x,{z_{ > k + 1}})}} \approx \frac{{R(x,{z_{ > k - 1}})}}{{R(x,{z_{ > k}})}}$, like $k=1$ or $k=0$.
> > Thus, $\mathcal{LLR}^{ada}$ can finally achieve the goal of adaptively enhancing the importance of some discriminative terms, like $\mathcal{LLR}^{>2}$, in the overall score function for OOD detection.
> >
> > It would help your understanding of different score methods with a numerical example in Table 12 and 13 in Appendix M.
> >
> > **5. Typos**
> >
> > Thanks for your notification, we have fixed these typos in the revision.
> >
> > **6. For "Tab. 2 left should probably read “FashionMNIST(in)/MNIST(out)"**
> >
> > Thanks, we have fixed it in the revision.
> >
> > **7. For "The exposition could be improved (e.g. l. 95 “at the cost of bringing heavy burdens”). It is not always clear in notation, if it is a definition or an implied equality (e.g. l. 120,188)."**
> >
> > Thanks, we have explained why Likelihood Regret will bring heavy computation burdens in the previous response to Q.~2, and modified the corresponding sentence as follows:
> >
> > "A pioneering VAE-based OOD detection method is Likelihood Regret (LRe) [23] calculated by iteratively fine-tuning the decoder parameters of VAE, which is time-consuming but achieves competitive performance in an unsupervised manner.''
> >
> > Thanks, the equations in Line 120 and 188 are both implied equations, such as $p\_\\theta(x\|z\_{&gt;k}) = \\int p\_\\theta(x, z\_{\\leq k}\|z\_{&gt;k}) dz\_{\\leq k}
> > = \\int p\_\\theta(x\| z\_{\\leq k}) p\_\\theta(z\_{\\leq k} \|z\_{&gt;k}) dz\_{\\leq k} = E\_{p\_\\theta(z\_{\\leq k}\|z\_{&gt; k})}\\left\[ p\_\\theta(x\|z\_{\\leq k}) \\right\].$
> >
> > **8. For "The acronyms used in Tab. 1 are not all self-explanatory, nor is it very clear which previous works specifically they come from. Please make it clearer in the revision" **
> >
> > Thanks for your notification, we have already cited these approaches in Section 4.1 and also added the citations of baselines in all Tables in the revision. For the details of these baselines, please refer to
> > the Appendix E named “Details of the Baselines” in our first submitted manuscript.
> >
> > **For Q1: "What is the OOD accuracy with the vanilla VAE loss (i.e. without the informative loss), but with the adaptive criterion"**
> >
> > Thanks for your suggestions, we have provided the additional experimental results of vanilla VAE with adaptive criterion as shown in Table  6, Table  7, and Table 12 in Appendix.
> > From the results, we can see that the developed Adaptive Likelihood Ratio can still outperform other OOD score functions on the vanilla VAE. Moreover, we provide an experimental analysis of the effect of different score methods on vanilla HVAE without informative loss in Appendix M.
> >
> > **For Q2: "Testing the approach on MNIST and CIFAR is great, but what about more natural images"**
> >
> > Thanks for your suggestions, due to limited time, we cannot provide experimental results on more natural images in the response.
> > But we promise that we will try to include these experiments during the discussion period if we can, and even evaluate our method on large-scale datasets in future work.

---

> > > ### Author Response · Authors · 2022-08-02
> > > **Response to Reviewer jV8d (3/3)**
> > >
> > > **For Q3: "How does the computational footprint change"**
> > >
> > > Take the vanilla VAE equipped with Likelihood Ratio as the baseline. For the space complexity, our method doesn't introduce any additional model parameters or memory cost. For the time complexity, compared to the baseline, our method requires additional $L-1$ times computation cost to calculate those expected log-likelihood terms in the loss function, specifically $\\frac{1}{L} \\sum\\nolimits\_{k=0}^{L-1} \\mathbb{E}\_{p\_\\theta(\mathbf{z}\_{\\leq k}\|\mathbf{z}\_{&gt; k})q\_\\phi(\mathbf{z}\_{&gt; k}\|\mathbf{x})} \\left\[ \\log p\_\\theta(\mathbf{x}\|\mathbf{z}\_{\\leq k}) \\right\]$, where $L$ denotes the number of layers and will be a relative small number in practice.
> > >
> > > **For limitations "The limitations in Sec. H are a bit unspecific ("additional computational burden"). The broader impact in Sec. I is perhaps a bit more technical than it should be."**
> > >
> > > Thanks for your suggestion, we have revised these in the revision.

---

> ### Author Response · Authors · 2022-08-08
> **Response to Reviewer jV8d for our promised items**
>
> Thanks again for your efforts in reviewing!
>
> As your awesome suggestion in "Questions" that
> **"Testing the approach on MNIST and CIFAR is great, but what about more natural images"**,
> we provide additional comparisons on more dataset pairs in appendix **P**, including Tiny-Imagenet, LFWPeople, Flower102, Places365, and Food101.

---

### Official Review · Reviewer_4FiY · 2022-07-14

**Rating:** 6
**Confidence:** 5
**Soundness:** 3 good
**Presentation:** 3 good
**Contribution:** 3 good

**Summary:**

This paper presents a hierarchical VAE (HVAE) for out-of-distribution (OOD) detection. Authors investigate the 'posterior collapse' problem with HVAE models and propose a solution to mitigate the by increasing the mutual information between the input and latent representations. Finally, an adaptive likelihood ratio-based measure is proposed for the HVAE models to detect OOD samples. The proposed approach is evaluated on benchmark datasets and the proposed approach outperforms related variants.

**Questions:**

Please address the comments in the weaknesses section, especially the novelty of the proposed approach with respect to existing approaches to preventing posterior collapse.

**Strengths And Weaknesses:**

Strengths:

1. Authors systematically explored the 'posterior collapse' problem with HVAE models. An interesting mitigation strategy is proposed by increasing the mutual information between the input and latent representations.

2. An adaptive likelihood ratio-based measure is proposed to distinguish the OOD sample using all layers of HVAE.

3. The draft is clearly written and easy to follow.

4. Authors experimentally evaluated various components of the approach justifying their contribution toward the final performance.


Weaknesses:

1. There are other approaches to prevent posterior collapse such as bidirectional inference [20] or oversmoothing VAE loss function. How does the proposed approach compare with these approaches and what is the fundamental difference? [20]  also considers skip connections.
Takida et al., "Preventing Posterior Collapse Induced by Oversmoothing in Gaussian VAE"

2. In table 1, the approaches are not cited.

3. Experiments are limited. Authors are encouraged to compare the results against the more recent state of the art such as

a. Weitang Liu, Xiaoyun Wang, John Owens, and Yixuan Li. Energy-based out-of-distribution detection. Advances in Neural Information Processing Systems, 2020.

b. Rui Huang, Andrew Geng, and Yixuan Li. On the importance of gradients for detecting distributional shifts in the wild, ArXiv, abs/2110.00218, 2021

c. Kaur, Ramneet, Susmit Jha, Anirban Roy, Sangdon Park, Edgar Dobriban, Oleg Sokolsky, and Insup Lee. "iDECODe: In-distribution equivariance for conformal out-of-distribution detection." AAAI (2022).

---

> ### Author Response · Authors · 2022-08-02
> **Response to Reviewer 4FiY (1/2)**
>
> Thanks for your effort in reviewing our paper!
>
> **For Q1: "Comparison and the fundamental difference with BIVA and Oversmoothing VAE.''**
>
> For BIVA [20], as discussed in the first paragraph of Section 3.3, it focuses on alleviating "posterior collapse'' by modifying the generative network structure of HVAE.
> Specifically, BIVA is characterized by a skip-connected generative model and an inference network formed by a bidirectional stochastic inference path, whose generative process forces the concatenation of latent variables $\\{\mathbf{z}\_k\\}\_{k=1}^{L}$ to be **physically** linked to the generated samples, potentially hurting the hierarchy of multiple latent representations.
> Without modifying the generative network structure of HVAE, the developed informative HVAE tries to introduce **virtual** skip-connection-liked structures into the objective function for training VAEs, specifically $E\_{q\_\\phi(\mathbf{z}\_{&gt; k}\|\mathbf{x})}\\left\[ \\log p\_\\theta(\mathbf{x}\|\mathbf{z}\_{&gt; k}) \\right\]$  terms to build straightforward connections between the observation $\mathbf{x}$ and latent variables $\mathbf{z}\_{>k}$ at higher layers,
> and its main idea can be applied to any existing hierarchical VAE, which is one of the main contributions of our work.
>
> For Oversmoothing VAE [1], its main idea is that an inappropriate variance $\sigma\_{\mathbf{x}}$ will cause the oversmoothness of the decoder and lead to ``posterior collapse'', where the $\sigma_{\mathbf{x}}$ is the variance parameter in the likelihood function $p_{\theta}(\mathbf{x}|\mathbf{z})=\mathcal{N}(\mathbf{x}|\mu_{\mathbf{x}}(\mathbf{z}),\sigma_{\mathbf{x}}^{2}\mathbf{I})$. Please note that, in Oversmoothing VAE, $\sigma_{\mathbf{x}}$ is not parameterized by networks, which is directly updated with a 1-dimensional value related to the training objective during the learning process instead. Thus, Oversmoothing VAE is developed to alleviate collapse specifically for the VAEs whose variances $\sigma_{\mathbf{x}}$ is fixed as a 1-dimensional constant parameter rather than being parameterized by networks like conventional VAEs, such as $\sigma_{\mathbf{x}}(\mathbf{z})$. However, in our paper, we adopt the most original settings for VAEs [2], where the variances $\sigma_{\mathbf{x}}$ is parameterized by fully-connected networks and has the same dimension as $\mathbf{x}$, where the likelihood function is $p_{\theta}(\mathbf{x}|\mathbf{z})=N(\mathbf{x}|\mu_{\mathbf{x}}(\mathbf{z}),\sigma_{\mathbf{x}}(\mathbf{z}))$.
> We note that the developed informative HVAE and Oversmoothing VAE alleviate "posterior collapse'' from exactly different perspectives, and we have also compared their effectiveness in the following experiments.
>
> Thanks for your bringing Oversmoothing VAE [1] to our eyes, and we have provided an additional comparison to demonstrate our method can beat it on unsupervised OOD detection, as the experimental results shown in Table 6 and Table 7 of Appendix J. For BIVA [20], we have already treated it as an important baseline in our experiments, and the comparison results can be found in the right part of Table 2 (termed as "HVK", since the HVK choose BIVA as their backbone in detecting SVHN as OOD), Table 3, Figure 3 (a~b) and Figure 4} of our first submitted manuscript. Besides, we also add an additional comparison between BIVA and other methods in  Table 6 and Table 7 of Appendix J.
>
> [1] Takida et al., "Preventing Posterior Collapse Induced by Oversmoothing in Gaussian VAE''.
>
> [2] Diederik P Kingma and Max Welling. ``Auto-encoding variational bayes''.
>
>
> **For Q2: "In table 1, the approaches are not cited.''**
>
> Thanks for your notification, we have already cited these approaches in Section 4.1 and also added the citations of baselines in all Tables in the revision. For the details of these baselines, please refer to the Appendix E named "Details of the Baselines'' in our first submitted manuscript.

---

> > ### Author Response · Authors · 2022-08-02
> > **Response to Reviewer 4FiY (2/2)**
> >
> > **For Q3: "Experiments are limited, Authors are encouraged to compare the results against the more recent state-of-the-art methods''**
> >
> > Thanks for your recommending these excellent works, and we have carefully read these papers.
> > We have also briefly summarized these papers as follows:
> > 1) Paper (a) develops an energy-based OOD method by replacing the softmax score with an energy one, but still utilizes the groundtruth class labels to estimate the corresponding energy;
> > 2) Paper (b) exploits the property of backpropagation gradients derived from the KL-divergence between the softmax output and a uniform distribution;
> > 3) Paper (c) is still a classifier-based method (labels are needed), which is developed based on the insight that DNN should be invariant to the transformation like data augmentation.
> >
> > In short, all of these methods are developed for supervised OOD detection and cannot be applied to unsupervised scenarios. We have discussed these supervised methods in the first paragraph of the Introduction).
> >
> > We emphasize that our work focuses on investigating purely unsupervised OOD detection methods, where the in-distribution data's class labels are not available and no prior knowledge of OOD data is allowed (no additional OOD datasets to help training and no assumption about the OOD data type), and have compared it with SOTA unsupervised OOD methods in the experiments, including HVK and a series of HVAE baselines.
> > Moreover, we have also included recent popular supervised methods termed "Label" and "Prior" as compared baselines in Table~1.
> > Considering that the aforementioned three papers [a] [b] [c] belong to the method category "Label", we have also included their OOD detection results in Table 1, such as paper [a] (termed as "EN") and paper [c] (termed as "iDE").
> >
> > Additionally, we have provided more experimental results in the revised Appendix, including
> > 1) comparison with more methods designed for alleviating posterior collapse like Oversmoothing VAE, Warm-up in Appendix J;
> > 2) comparison with more non-VAE baselines for unsupervised OOD detection like flow-based models in Appendix K;
> > 3) comparison with different score methods in Appendix L and M;
> > 4) t-sne visualization of hierarchical latent representations in Appendix N;
> > 5) measure of reconstruction quality with partial generative models $p_\theta(\mathbf{x}|\mathbf{z}_{>k})$ and visualization of data samples generated from the prior distribution in Appendix O.
> > --------------------------------------------------------------------------------------
> > 6) **(update on 8 August) we add an additional comparison on more dataset pairs in appendix P, including Tiny-Imagenet, LFWPeople, Flower102, Places365, and Food101.**

---

> ### Author Response · Authors · 2022-08-07
> **Response to Reviewer 4FiY (Further Discussion)**
>
> Dear Reviewer 4FiY,
>
> Thanks a lot for your valuable comments to improve this paper! Are there unclear explanations here based on our response? We are willing to further clarify them and have a discussion with you in the following days!
>
> Best regards,
>
> Authors

---

> ### Author Response · Authors · 2022-08-09
> **Further Discussion**
>
> Dear Reviewer 4FiY:
>
> Thanks again for your effort in reviewing our paper and give us a great chance to improve the quality of this paper .
>
> Considering that the discussion period is coming to an end, we would like to know if you have any other questions about our paper, and we are still glad to have a discussion with you in the limited time.
>
> Sorry for disturbing you again and again, we only want to let you know your decision is quite important to us.
>
> Sincerely
>
> Authors

---

### Official Review · Reviewer_1sqv · 2022-08-02

**Rating:** 7
**Confidence:** 3
**Soundness:** 3 good
**Presentation:** 3 good
**Contribution:** 3 good

**Summary:**

The paper investigates the problem on "posterior collapse" in hierarchical variational autoencoders (HVAE), and provides a theoretical explanation for why this occurs during training based on the ELBO lower bound. It further discusses why posterior collapse can affect the OOD detection performance of the HVAE model. Based on these insights, the paper proposes to enhance the connection (dependence) between the input an its multilayer stochastic latent representations based on an informative HVAE training objective. It also proposes an adaptive likelihood ratio score for detecting OOD inputs, which enhances the separation in in-distribution and OOD inputs, and does not depend on the specific choice of higher-level latent layer representations used.

**Questions:**

While the paper is technically strong, it is a bit hard to follow and there are some logical jumps that are not obvious. Overall, the presentation of ideas could be improved.

1. It is mentioned that the proposed method is completely unsupervised, i.e., it does not require labels for the in-distribution data, nor does it require auxiliary OOD data for the detection algorithm. The proposed method is different from many conventional OOD detection methods in that it does not depend on a classification model (DNN) for its scoring. In this sense, it is more like an anomaly detection method.
Can the authors comment on whether the proposed method can be improved by utilizing the class labels (if available) by modeling the class-conditional distributions of $x$?

1. The paper does not discuss a number of prior works on OOD detection in section 2. These include methods not based on deep generative models such as maximum softmax probability [1], Generalized ODIN [2], Deep Mahalanobis [3], Energy-based [4], ReAct [5] etc.

1. On line 89, it is mentioned that likelihood methods based on generative models always assign a higher likelihood to OOD inputs compared to ID inputs. While this happens in some cases, it is not always true.

1. In equation (3), it should be clarified that $z_{L+1} := x$ as a special case. Otherwise the dependence on $x$ is not clear in the second term.

1. Please provide a discussion on the log-likelihood ratio score in Eqn. (5).

1. On line 161, it should be $q_\phi$ and not $q_\theta$.

1. On line 185, it is mentioned that the entropy of the marginal distribution on $x$ is a positive constant. This does not have to be the case for continuous $x \in \mathbb{R}^d$. If so, is the lower bound on the mutual information correct?

1. The informative loss proposed in Eqn. (8) and its lower bound could be discussed in more detail.

1. It is not obvious how the authors arrive at the proposed adaptive log-likelihood score. Please provide some intuition about why this is formulated as a weighted difference of log-likelihood ratios. Could this be arrived at in a more principled way? Why does it enhance the separation between ID and OOD inputs?

1.  Minor: On lines 241 - 243, it is mentioned that the metrics are threshold independent, but the `FPR80` metric does depend on the threshold.

1.  The paper compares with different categories of OOD detection methods, but some of them are not defined in the main paper. This makes it a big vague to read the tables and figures.

1. In Table 1, why is the AUPRC not reported? This metric is sometimes more effective is capturing the separation between the OOD and in-distribution scores, especially at low FPR.

1. In Table 1, it is not clear why some of the baseline methods are missing for the CIFAR10 / SVHN datasets. The format is a bit confusing to follow.

1. In Table 2, why is SVHN used as the OOD dataset for FashionMNIST, whereas MNIST is used as the OOD dataset in Table 1? More complete results could be provided in the Appendix.

1. In Figure 3, could the authors also report the area under the ROC curves, maybe as part of the legend.


### References

[1] Hendrycks, Dan, and Kevin Gimpel. "A baseline for detecting misclassified and out-of-distribution examples in neural networks." arXiv preprint arXiv:1610.02136 (2016).

[2] Hsu, Yen-Chang, et al. "Generalized odin: Detecting out-of-distribution image without learning from out-of-distribution data." Proceedings of the IEEE/CVF Conference on Computer Vision and Pattern Recognition. 2020.

[3] Lee, Kimin, et al. "A simple unified framework for detecting out-of-distribution samples and adversarial attacks." Advances in neural information processing systems 31 (2018).

[4] Liu, Weitang, et al. "Energy-based out-of-distribution detection." Advances in Neural Information Processing Systems 33 (2020): 21464-21475.

[5] Sun, Yiyou, Chuan Guo, and Yixuan Li. "React: Out-of-distribution detection with rectified activations." Advances in Neural Information Processing Systems 34 (2021): 144-157.



**Limitations:**

The discussion of limitations and broader impact in Appendix H and I is adequate.

**Strengths And Weaknesses:**

Strengths:

The paper addresses an important problem of posterior collapse observed in hierarchical VAEs, which significantly limits the ability of the model to be used for OOD detection. The motivation and background on posterior collapse based on mutual information is interesting, and there is adequate discussion of related work on the problem. Building on the insights, the paper proposes a novel training objective (informative HVAE) which alleviates the issue of  posterior collapse by enhancing the dependence between the input and the higher level latent variables. The experiments are fairly extensive and compare the OOD detection performance of the proposed method with a number of baselines from different categories of OOD detection.

Weaknesses:

- The technical details are hard to follow in some places and lacks enough discussion. For instance, the likelihood-ratio score in Eq. (6) and its approximation are not discussed clearly. Same comment for the informative HVAE loss in Eqs. (8) and (9).

- The paper compares with several baselines, but they are not clearly defined in the main paper (please questions on the experiments).

- In the experiments, only two in-distribution datasets are evaluated, while it is common to evaluate on more ID/OOD dataset pairs. Some additional results, including an ablation study on the adaptive log-likelihood score, could be provided in the appendix to make the evaluation stronger.

- A number of typos and grammatical issues, which could be easily fixed by proof-reading.

---

> ### Author Response · Authors · 2022-08-02
> **Response to Reviewer 1sqv (1/2)**
>
> Thanks for your effort in reviewing the paper!
>
> **For  weakness 1**
>
> Thanks, we have revised our paper and provided more comments so that our method can be intuitively understood. We have given an intuitive explanation of the likelihood-ratio score in our revision (Section 3.2) and left the derivation details in Appendix B. We have also added more comments to illustrate Eq. (8) and Eq. (9) in the revision.
>
> **For weakness 2**
>
> Thanks for your constructive suggestion.
> We have discussed these baselines in Section 4.1 and left more details to Appendix E named "Details of the Baselines''.
> We promise that we will try to move the definitions of these baselines in future revision.
>
> **For weakness 3**
>
> Thanks for your constructive suggestion.
> Additionally, we have provided more experimental results in the revised Appendix, including
> 1) comparison with more methods designed for alleviating posterior collapse like Oversmoothing VAE, Warm-up in Appendix J;
> 2) comparison with more non-VAE baselines for unsupervised OOD detection like flow-based model in Appendix K;
> 3) comparison with different score methods in Appendix L and M;
> 4) t-sne visualization of hierarchical latent representations in Appendix N;
> 5) measure of reconstruction quality with partial generative models $p_\theta(x|z_\{>k\})$ and visualization of data samples generated from the prior distribution in Appendix O.
>
> **For weakness 4**
>
> Thanks, we have fixed the typos and grammatical issues you mentioned, and will further improve the quality of the paper.
>
> **For Question 1**
>
> Thanks for your awesome suggestion for utilizing the class label into the VAE for OOD detection. A very straightforward idea is that the label could be used to guide the learning of latent space, e.g., we could model the posterior distribution $q(z|x)$ as a Gaussian-Mixture-Model (GMM) $q(z|x) = \sum_{i=1}^{\operatorname{C}} w_i \mathcal{N}(z_i|\mu_i(x), \sigma_i(x))$, where $C$ is the number of classes and the label could be used to guide the learning of the coefficient $w_i$ of the GMM, which indicates the probability for each classes conditioned on $x$. We think this idea could further improve the performance of the OOD detection with the help of class labels during training. We are trying to work in this direction.
>
> **For Question 2**
>
> Thanks for your bringing these excellent works to our eyes. Due to the time limitation, we can only capture the main idea of these works and find them all belong to supervised OOD detection methods. Thus, we choose to only discuss these work in the first paragraph of the introduction in this revision, and will carefully study them in the future.
>
> **For Question 3**
>
> Yes, it is correct that the mentioned phenomenon will not always happen. The point we want to highlight is that our method can successfully deal with these cases, and also still work well on daily scenarios (the mentioned phenomenon does not happen).
>
> **For Question 4**
>
> Thanks for your careful review! We have clarified $\mathbf{z}_{L+1}:=\mathbf{x}$ in our revision.
>
> **For Question 5**
>
> Thanks, we have given an intuitive explanation of the likelihood-ratio score in our revision (Section 3.2), and left the derivation details in Appendix B.
>
> **For Question 6**
>
> Thanks, we have fixed this typo!
>
> **For Question 7**
>
> Thanks for your notification. Actually, $H_p(x)$ is the entropy conditioned on the **true** distribution $p(x)$ of data $x$, and a reasonable prior assumption for it is that it should obviously not be the extreme case like the impulse function $\delta(x)$ but a smoother one, where the value for it could be bounded into 0~1 in our setting. Thus, in this setting, the entropy $H_p(x)$ is a non-negative value.
>
> **For Question 8**
>
> Thanks, we have added more comments to illustrate Eq. (8) and Eq. (9) in the revision.

---

> > ### Author Response · Authors · 2022-08-02
> > **Response to Reviewer 1sqv (2/2)**
> >
> > **For Question 9**
> >
> > Thanks for your suggestions. We have revised Section 3.4 and give a brief explanation here.
> >
> > Firstly, we need to point out that, for the likelihood-ratio score function $\mathcal{LLR}^{>k}$ [19], cherry picking the hyperparameter $k$ on testing OOD samples is unreasonable for unsupervised OOD detection, but an inappropriate choice of $k$ will bring negative impact on the performance.
> > Thus, the intuition of designing $\mathcal{LLR}^{ada}$ is to move beyond the choose of $k$ but adaptively enhance the importance of some discriminative terms, like $\mathcal{LLR}^{>2}$, in the overall score function for OOD detection.
> > With $\mathcal{R}(x, z_{>k})$ to measure the relevance between $x$ and $z_{>k}$, we find that the adaptive weight $\frac{\mathcal{R}(x, z_{>k-1})}{\mathcal{R}(x, z_{>k})}$ will be relatively large when the data information drop rapidly at the current hidden layer, like $k=2$, which naturally meets our requirements for designing $\mathcal{LLR}^{ada}$.
> >
> > Through combining $\frac{\mathcal{R}(x, z_{>k-1})}{\mathcal{R}(x, z_{>k})}$ and $\mathcal{LLR}^{>k}$, there could be several ways to design the final score function. The reason why we choose $\mathcal{LLR}^{ada}$ (a weighted difference of log-likelihood ratios) in Eq. (10) is that it can numerically omit some terms that occur ``posterior collapse'', as discussed in Q.4 to review jv8d.
> >
> > Then, we believe that there will be other more principled or effective ways to redesign the score function through combining $\frac{\mathcal{R}(x, z_{>k-1})}{\mathcal{R}(x, z_{>k})}$ and $\mathcal{LLR}^{>k}$, and we are working on this.
> >
> >
> > **For Question 10**
> >
> > Thanks for your careful review, we will revise it.
> >
> > **For Question 11**
> >
> > Thanks for your notification, we have already cited these approaches in Section 4.1 and also added the citations of baselines in all Tables in the revision. For the details of these baselines, please refer to Appendix E named ``Details of the Baselines'' in our first submitted manuscript.
> >
> > **For Question 12**
> >
> > Thanks for the suggestion! We have added more comparisons under these metrics (AUPRC, AUROC, and FPR80) with non-VAE methods in  Appendix K and other methods designed for alleviating "posterior collapse'' in Appendix J.
> >
> > **For Question 13**
> >
> > Some of the results of the baseline shown in Table 1 are directly cited from their original papers, and we only report the results under the same experimental setting as ours. We have also provided additional comparisons of the baselines in Appendix K and J.
> >
> > **For Question 14**
> >
> > Sorry, this is a typo and we have fixed it in the revision.
> >
> > **For Question 15**
> >
> > Thanks for your suggestion. We have reported the area under the ROC curves in Fig. 3 of our revision.;

---

> ### Author Response · Authors · 2022-08-08
> **Response to the Reviewer 1sqv for our promised items**
>
> Thanks again for your efforts in reviewing! We carefully read the papers you recommended and we give an additional brief summary of them below:
>
> 1) For paper [1], it is based on the belief that correctly classified examples tend to have greater maximum softmax classification probabilities than out-of-distribution examples.
>
> 2) For paper [2], it is an extension of ODIN, and it still relies on the classification confidence score.
>
> 3) For paper [3], it is designed for pre-trained softmax neural classifiers.
>
> 4) For paper [4], it replaces the softmax score with the energy score for pre-trained classifiers, but it still needs the label to estimate the energy.
>
> 5) For paper [5], it applies rectified activation operation to the penultimate layer of a classifier.
>
> In short, all these papers for OOD detection still require the category labels of in-distribution data samples, which is not applicable in our setting.
>
>
> We note that, although the OOD detection methods especially with the help of in-distribution data labels have been well studied,
> the purely unsupervised OOD detection (no labels and no prior assumption for the OOD data) is still rarely investigated
> due to the challenging setting.
> However, unsupervised OOD detection is suitable and practical for more scenarios, especially in cases where labels are not available.
>
>
> Besides, as your awesome suggestion in "Weakness" that
> **"only two in-distribution datasets are evaluated, while it is common to evaluate on more ID/OOD dataset pairs"**,
> we provide addtional comparisons on more dataset pairs in **appendix P**, including Tiny-Imagenet, LFWPeople, Flower102, Places365, and Food101.

---

> > ### Comment · Reviewer_1sqv · 2022-08-08
> > **Follow-up to author response**
> >
> > Thank you for the detailed response and efforts on the revised submission. My concerns have been mostly addressed, and I am willing to increase my score to 7.
> >
> > I wanted to clarify one point about the entropy $\mathcal{H}_p(x)$ in Eqn. (8) of the revised paper. Consider for example the case where $x$ follows a standard Gaussian distribution, and for simplicity let it be univariate. In this case, the entropy of $x$ would be $1/2 + 1/2 \log(2 \pi \sigma^2)$. This can be negative when $\sigma^2 < \frac{1}{2 \pi e}$. Similarly if $x$ is multivariate Gaussian, the entropy of $x$ can be negative when the determinant of its covariance satisfies $|\Sigma| < e^{-d (1 + log(2 \pi))}$, where $d$ is the dimension.
> >
> > Given that the entropy of $x$ can be negative for some distributions, how valid is the inequality  $I_p(x, z_{>k}) \geq H_{p,q}(x | z_{>k})$ (on line 189)? Is this a requirement satisfied in practice when the empirical estimate of the entropy are used?

---

> > > ### Author Response · Authors · 2022-08-09
> > > **Further response to Reviewer 1sqv**
> > >
> > > Thanks for your effort in checking our paper! We agree with your comment that the inequality $I_p(\mathbf{x}, \mathbf{z_{>k}}) \geq H_\{p,q\}(\mathbf{x}|\mathbf{z_\{>k\}}) $ will not hold when the entropy $H_{p}(\mathbf{x})$ is negative in the special cases, though which is not easy to happen when $d$ is a large value like $28\times28\times1$ in MNIST (i.e., $|\Sigma|< e^{-28\times28\times(1+\log 2\pi)}$).
> > >
> > > Given the cases where $H_{p}(\mathbf{x})$ is negative, however, please note that $H_{p}(\mathbf{x})$ is a constant, which is the expectation only related to the **true** distribution $p(x)$ of the data $x$ and does not change along with the training.
> > > Thus, there still exists $I_p(\mathbf{x}, \mathbf{z_\{>k\}}) = E_{p(\mathbf{x})p_\theta(\mathbf{z_\{>k\}}|\mathbf{x})}\log p_\theta(\mathbf{x}|\mathbf{z_\{>k\}}) + H_{p}(\mathbf{x}) \propto E_{p(\mathbf{x})p_\theta(\mathbf{z_\{>k\}}|\mathbf{x})}\log p_\theta(\mathbf{x}|\mathbf{z_\{>k\}})$ in Eq. (8), and this item can be approximated by $H_\{p,q\}(\mathbf{x}|\mathbf{z_\{>k\}})=E_{p(\mathbf{x})q_\phi(\mathbf{z_\{>k\}}|\mathbf{x})}\log p_\theta(\mathbf{x}|\mathbf{z_\{>k\}})$, which indicates that our approach can still work well even if the entropy $H_{p}(\mathbf{x})$ is a negative constant.
> > >
> > > In other words, the optimization direction of maximizing $H_{p,q}(\mathbf{x}|\mathbf{z_\{>k\}})=E_{p(\mathbf{x})q_\phi(\mathbf{z_\{>k\}}|\mathbf{x})}\log p_\theta(\mathbf{x}|\mathbf{z_\{>k\}})$ is consistent with maximizing  $I_p(\mathbf{x}, \mathbf{z_\{>k\}})$ even if the entropy $H_{p}(\mathbf{x})$ is a negative constant in some extreme data distributions.
> > >
> > > We thank you again for your careful review and have revised our paper in this submission.

---

> > > > ### Comment · Reviewer_1sqv · 2022-08-09
> > > > **Thanks for the clarification**
> > > >
> > > > This is now clear to me.

---

> > > > > ### Author Response · Authors · 2022-08-09
> > > > > **About the score**
> > > > >
> > > > > Thanks for your effort in reviewing our paper.
> > > > >
> > > > > One more thing, it seems that you haven't updated the socre, which is quite important to us.
> > > > >
> > > > > Best wishes
> > > > > Authors

---

> > > > > > ### Comment · Reviewer_1sqv · 2022-08-09
> > > > > > **Score updated**
> > > > > >
> > > > > > The score should be updated to 7. Shows on my end.

---

> > > > > > > ### Author Response · Authors · 2022-08-09
> > > > > > > **Thanks!**
> > > > > > >
> > > > > > > Best wishes!
> > > > > > >
> > > > > > > Authors

---

### Meta-Review · Area_Chair_JDNH · 2022-08-27

**Recommendation:** Accept
**Confidence:** Certain

**Metareview:**

This paper studies unsupervised out-of-distribution detection based on hierarchical VAE models. In particular, it (1) investigates the posterior collapse issue, (2) proposes a training procedure by increasing the mutual information between the input and latent representations, and (3) proposes an adaptive likelihood ratio score for detecting OOD inputs. Multiple reviewers found the method interesting and technically sound.

Post rebuttal, all reviewers unanimously supported this paper positively. The contribution and insights presented in this paper will be valuable for the OOD detection community. The AC recommends acceptance.

Please incorporate the reviewer's requested discussions (e.g. computational footprint) in the final version. Several published papers in the reference sections are in arXiv format, which necessitates proper citations in camera ready.







**Award:**

No

---

### Decision · Program_Chairs · 2022-09-14

Accept